# Point Cloud Matters: Rethinking the Impact of Different Observation Spaces on Robot Learning

**Haoyi Zhu**[12]   **Yating Wang**[23]   **Di Huang**[2]   **Weicai Ye**[24]   **Wanli Ouyang**[2]   **Tong He**[2†]

[1]University of Science and Technology of China   [2]Shanghai Artificial Intelligence Laboratory
[3]Northwestern Polytechnical University   [4]Zhejiang University
hyizhu1108@gmail.com
{wangyating,huangdi,yeweicai,ouyangwanli,hetong}@pjlab.org.cn
[†]Corresponding Author
https://github.com/HaoyiZhu/PointCloudMatters

## Abstract

In robot learning, the observation space is crucial due to the distinct characteristics of different modalities, which can potentially become a bottleneck alongside policy design. In this study, we explore the influence of various observation spaces on robot learning, focusing on three predominant modalities: RGB, RGB-D, and point cloud. We introduce OBSBench, a benchmark comprising two simulators and 125 tasks, along with standardized pipelines for various encoders and policy baselines. Extensive experiments on diverse contact-rich manipulation tasks reveal a notable trend: point cloud-based methods, even those with the simplest designs, frequently outperform their RGB and RGB-D counterparts. This trend persists in both scenarios: training from scratch and utilizing pre-training. Furthermore, our findings demonstrate that point cloud observations often yield better policy performance and significantly stronger generalization capabilities across various geometric and visual conditions. These outcomes suggest that the 3D point cloud is a valuable observation modality for intricate robotic tasks. We also suggest that incorporating both appearance and coordinate information can enhance the performance of point cloud methods. We hope our work provides valuable insights and guidance for designing more generalizable and robust robotic models.

## 1   Introduction

The evolution of robot learning has been profoundly influenced by the integration of visual observations, which enable robots to perceive and interact with complex environments. A key challenge faced by contemporary robot models is their limited generalization ability, particularly in dynamic and intricate settings, due to partial observability.

Researchers in robot learning primarily focus on policy design [4, 9, 68, 87]. However, these policies depend on inputs derived from estimated world states or features. Consequently, different observation spaces, such as RGB, RGB-D, or point clouds, can significantly influence robotic performance. This makes the observation space a potential bottleneck, impacting the robot's ability to generalize and perform effectively in various environments.

Commonly, robotic vision has predominantly utilized 2D images [64, 58, 51, 54, 52, 4, 87, 9] for their simplicity and the considerable advancements in 2D foundation models [33, 3, 63, 34]. Despite their ubiquity, RGB images often fall short of accurately capturing the 3D structure of environments, essential for precise action execution. These methods also exhibit vulnerable generalization to changes such as lighting conditions and camera viewpoints, owing to their reliance on appearance.

38th Conference on Neural Information Processing Systems (NeurIPS 2024) Track on Datasets and Benchmarks.

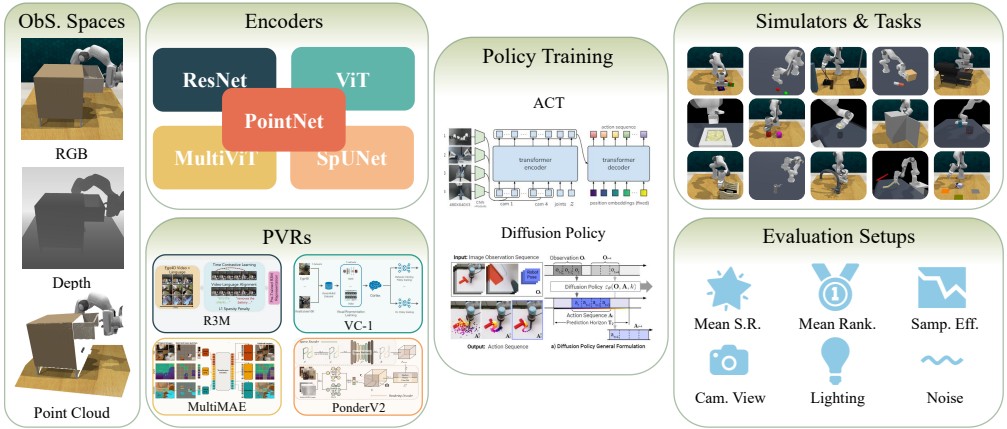

Figure 1: **Overview of this work.** We examine the impact of various observation spaces, specifically RGB, RGB-D, and point clouds, on robot learning. We develop OBSBench, a benchmark with standardized pipelines that include various encoders, PVRs, policies, simulators, evaluation settings, *etc*. Based on OBSBench, we conduct a series of empirical studies on observation spaces.

Given that robot action spaces are 3D, integrating 3D information into observation spaces appears inherently reasonable, as done in methods like CLIPort [67] , Perveiver-Actor [68] and ACT3D [26].

The suitability of different observation modalities for robotic tasks remains unexplored due to the lack of a unified comparative framework in the existing literature. Therefore, we introduce OBSBench, a benchmark built upon two modern robot simulators: ManiSkill2 [30] and RLBench [38]. OBSBench comprises 125 contact-rich tasks, each with ground-truth demonstrations available in all modalities. As far as we know, this is the first extensive benchmark for assessing and comparing the effectiveness of various observation spaces in robot learning.

Based on OBSBench, we conduct extensive experiments on different observation modalities across 19 diverse representative tasks. We first evaluate each observation space under identical settings, differing only in input modality and corresponding encoders. We choose backbones with similar model sizes. Recognizing the growing prevalence and varied efficacy of pre-trained visual representations (PVRs) across different modalities, we also investigate the performance of state-of-the-art PVRs within each observation space. Additionally, we also focus on the zero-shot generalization capabilities regarding camera viewpoints, lighting conditions, visual appearance, and sampling efficiency. Finally, we explore how to better utilize point clouds with different design choices such as sampling strategies and geometric information usage.

To the best of our knowledge, our OBSBench is the first to undertake such an extensive comparison of different observation spaces. Our non-trivial and insightful findings can be summarized as follows:

- Point clouds have emerged as a promising observation modality for robot learning, showing the highest mean success rate and best mean rank, whether trained from scratch or with pre-training. Notably, point cloud PVRs utilize significantly less pre-training data. This trend is also observed across different policy implementations.

- An explicit 3D representation is crucial for optimal performance. Utilizing 2D depth maps often shows constrained model performance, whether depth alone is used, RGB-D images are stacked channel-wise, or RGB and depth images are processed separately. Furthermore, adopting a seemingly compromised pointmap format also manifests limited performance.

- Point clouds demonstrate significantly greater robustness to variations in camera views and visual changes.

- We point out several key designs for improved accuracy and robustness with point cloud, such as feature sampling strategy and incorporating both color and coordinate features.

## 2 Background

**Problem Formulation.** The fundamental aim of robot learning is to develop a policy, $\pi(\cdot|o_t, \tau)$, which derives actions from visual observations. Here, $o_t \in \mathcal{O}$ is the observation at time $t$, and $\tau$ represents an optional task-specific target. The robot, guided by this policy, executes an action $a_t \in \mathcal{A}$, generating new observations and receiving a binary reward $r \in 0, 1$ at each episode's end. Our goal is to maximize expected rewards, considering task distribution, initial observations, and transition dynamics.

**Observation Space.** We analyze in detail the observation spaces $\mathcal{O}$, emphasizing that observations $o_t$ are projections of the real world states $s_t \in \mathcal{S}$ via different sensors $h(\cdot) : \mathcal{S} \rightarrow \mathcal{O}$. Our policy, therefore, is $\pi_h(\cdot|o_t = h(s_t), \tau)$. We explore the diversity of $h$, focusing on three observation spaces: RGB images, RGB-D images, and point clouds.

**Behavior Cloning.** We adopt behavior cloning [60, 45, 86] for policy learning due to its simplicity and universality. This method trains $\pi$ on a dataset $\mathcal{D}$ of successful demonstrations. The objective is to align the robot's actions with these demonstrations by optimizing $\pi$ to minimize the negative log-likelihood of the actions based on the observations and goal conditions.

## 3 OBSBench

Previous literature lacks a comprehensive and fair comparison of different observation spaces due to the varied settings, such as policy networks, datasets, augmentation methods, and training techniques. To address this, we have collected a detailed benchmark called **OBSBench**. We implemented standardized pipelines with a series of different baselines to establish a coherent framework for examining the impact of different observation spaces in robot learning. The codebase is built using Hydra [76] and PyTorch Lightning [21], making experiments easily configurable and flexible.

**Simulators and Tasks.** In recent years, many advanced simulators have emerged, facilitating reproducible and efficient robot benchmarking. Our study employs two well-known simulators, namely ManiSkill2 [30] and RLBench [38]. These simulators use different physics engines —SAPIEN and CoppeliaSim —making them sufficiently representative. For the ManiSkill2 simulator, we support 12 rigid body tasks and 5 soft body tasks from the official ManiSkill2 Challenge. For the RLBench simulator, we support all official 108 tasks. For all tasks in OBSBench, we provide demonstration trajectories with different observation modality replays. The advanced capabilities of these two simulators also allow for the convenient generation of more customized tasks. Detailed descriptions and examples of the tasks can be found in Appendix A.

**Encoders for different observation spaces.** We primarily consider three common visual modalities: RGB images, RGB-D images, and point clouds. For each modality, we offer standardized implementations of several commonly used encoders. For RGB images, we utilize • **ResNet** [32] and • **ViT** [18]. For RGB-D images, we use channel-wise stacked (*i.e.* the input images have 4 channels) • **ResNet** [32] and • **ViT** [18]. Additionally, we use • **MultiViT** [1], a Vision Transformer variant designed specifically for multi-modal inputs with distinct projection layers, which can effectively integrate RGB with depth information. For point clouds, we select • **PointNet** [62] and **Sparse-UNet (• SpUNet)** [14]. • PointNet is a popular point-based network, while • SpUNet is a sparse convolutional network widely adopted in the 3D vision community for point cloud perception tasks.

**Feature Extraction.** To use any observations, we must extract features from encoders to feed into our policy networks. We adopt commonly used settings for feature extraction: • ResNet uses final layer features, while • ViT and • MultiViT employ the `[CLS]` token. Regarding point cloud baselines, we use farthest point sampling (FPS) [20, 62] and K-nearest neighborhood (KNN) [19, 15]. Specifically, on the final 3D sparse convolutional feature map, we first use FPS to select $S$ seed points, then employ KNN to form $S$ clusters around the seeds. These clusters undergo a linear projection and pooling layer, resulting in $S$ features as inputs for the policy network. We aim to apply simple and common methods to facilitate a fair comparison.

**Policy Networks.** We implement two state-of-the-art policy networks: Action Chunking Transformer (ACT) [87] and Diffusion Policy (DP) [9]. ACT [87] models behavior cloning using a conditional VAE [70] while DP utilizes a diffusion process to model the observation and action spaces. They

both have demonstrated remarkable success in a variety of fine-grained manipulation tasks, both in simulated and real-world settings.

For more details, see Appendix B.

# 4 Experiments

We conduct empirical experiments on 19 selected tasks from OBSBench, considering computational resource constraints. The tasks are chosen to be diverse and representative. Details of the selected tasks are provided in Appendix A. We first introduce the evaluation metrics used in the experiments, followed by an empirical analysis of the results on OBSBench. The detailed experimental setups are described in Appendix C. Our experiments aim to address the following research questions:

**Q1:** How do varying observation spaces influence robot learning performance?
**Q2:** What is the performance impact of pre-trained visual representations (PVRs)?
**Q3:** How are the zero-shot generalization capabilities across observation spaces?
**Q4:** What is the sample efficiency across observation spaces?
**Q5:** How do different design decisions influence point cloud performance?

## 4.1 Evaluation Metrics

For each task, we employ **Success Rate (S.R.)** to evaluate the performance of each method. In addition, we adopt the evaluation methodology from VC-1 [52] encompassing two key metrics: **Mean S.R.** and **Mean Rank**. *Mean S.R.* calculates the average success rate across all tasks, providing an overall performance indicator. *Mean Rank*, on the other hand, involves ranking each method based on their success rate for each task and then averaging these rankings across all tasks. This metric offers insights into the relative performance of methods across diverse tasks.

## 4.2 Study on performance of different observations with and without pre-training (Q1, Q2)

We examine the task performance of different observation spaces across all tasks. The model parameters and the size of the pre-training data are provided in Tab. 1, with the results presented in Tab. 2. Each encoder is trained from scratch on each task using identical training data, pre-processing pipelines, policy architectures, and hyper-parameters, with the only variable being the input observation modalities, ensuring a fair comparison. To explore the impact of the depth modality, we conduct depth-only experiments. However, depth-only experiments are not feasible on RLBench, as RLBench evaluates models

Table 1: **Overview of encoders and corresponding PVRs.** #Params denotes the number of model parameters while #Data represents the number of images or point clouds during pre-training.

| Obs. Space | Encoder | #Params | PVR | #Data |
|---|---|---|---|---|
| RGB(-D) | ● ResNet50 | 23.5M | ○ R3M | 5M |
| | ● ViT-B | 85.8M | ○ VC-1 | 5.6M |
| RGB-D | ● MultiViT-B | 86.1M | ○ MultiMAE | 1.28M |
| Point Cloud | ● SpUNet34 | 39.2M | ○ PonderV2 | 4.5K |
| | ● PointNet | 0.14 M | - | - |

with varying color variations. Additionally, we analyze the effectiveness of pre-training on different observation spaces using state-of-the-art pre-trained visual representations (PVRs) for each encoder. The results of the PVRs are displayed in Tab. 3. Detailed information about each encoder and PVR can be found in Appendix B.

*Finding 1:* We observe that using a point cloud encoder results in the highest mean success rate and the best mean rank. **Point cloud methods consistently outperform other modalities**, securing the first or second rank across all 19 tasks, whether employing ACT policy or diffusion policy. Specifically, in terms of mean success rate, ● SpUNet and ● PointNet outperform the best other modality **by 53.85% and 76.92%**, respectively, when using the diffusion policy. This demonstrates the robustness and superiority of point cloud representations.

*Finding 2:* Despite providing geometric information, *the depth modality generally **degrades** performance across all settings*. This includes scenarios where only depth data is used, where RGB-D images are stacked channel-wise, or when using specialized architectures like ● MultiViT to process RGB and depth information separately. Although the RGB-D version of ● ResNet has a slightly better mean success rate than the RGB version when using ACT, it performs significantly worse on 7

Table 2: **Results on different observation spaces with different policies.** All encoders are trained from scratch. The best performance is **bolded** and the second best performance is underlined.

| Tasks | | ACT Policy | | | | | | | | |
| --- | --- | --- | --- | --- | --- | --- | --- | --- | --- | --- |
| | | RGB | | RGB-D | | | Point Cloud | | Depth Only | |
| | | ● ResNet | ● ViT | ● ResNet | ● ViT | ● MultiViT | ● SpUNet | ● PointNet | ● ResNet | ● ViT |
| PickCube | | 0.60 | 0.14 | 0.75 | 0.03 | 0.04 | 0.74 | **0.84** | 0.05 | 0.01 |
| StackCube | | 0.32 | 0.00 | 0.17 | 0.00 | 0.00 | 0.22 | **0.35** | 0.00 | 0.00 |
| TurnFaucet | | **0.49** | 0.27 | 0.00 | 0.06 | 0.35 | 0.39 | 0.00 | 0.41 | 0.00 |
| Peg- | Grasp | 0.73 | 0.36 | 0.73 | 0.03 | 0.16 | **0.81** | 0.77 | 0.07 | 0.01 |
| Insertion- | Align | 0.18 | 0.02 | 0.06 | 0.00 | 0.01 | 0.28 | **0.40** | 0.00 | 0.00 |
| Side | Insert | **0.01** | 0.00 | 0.00 | 0.00 | 0.00 | **0.01** | 0.01 | 0.00 | 0.00 |
| Excavate | | 0.02 | 0.00 | 0.02 | 0.14 | 0.00 | 0.03 | 0.27 | **0.29** | 0.00 |
| Hang | | **0.86** | 0.80 | 0.81 | 0.00 | 0.84 | 0.84 | 0.83 | 0.79 | 0.41 |
| Pour | | 0.07 | 0.00 | 0.01 | 0.00 | 0.00 | 0.10 | **0.14** | 0.00 | 0.00 |
| Fill | | 0.79 | 0.30 | 0.60 | 0.79 | 0.76 | 0.66 | **0.91** | 0.51 | 0.00 |
| open drawer | | 0.00 | 0.16 | 0.08 | 0.00 | 0.20 | **0.44** | 0.00 | - | - |
| sweep to | | 0.72 | 0.80 | **1.00** | 0.92 | 0.68 | 0.90 | **1.00** | - | - |
| meat off grill | | 0.24 | 0.16 | 0.36 | 0.08 | 0.00 | **0.72** | 0.44 | - | - |
| turn tap | | 0.00 | 0.00 | 0.00 | 0.00 | 0.00 | 0.00 | **0.04** | - | - |
| reach and drag | | 0.32 | 0.28 | **0.60** | **0.60** | 0.04 | 0.20 | **0.60** | - | - |
| put money | | 0.60 | 0.76 | **0.84** | 0.04 | 0.28 | 0.60 | 0.32 | - | - |
| push buttons | | 0.12 | 0.40 | 0.28 | 0.08 | 0.14 | 0.00 | **0.52** | - | - |
| close jar | | 0.04 | 0.00 | **0.16** | 0.00 | 0.00 | 0.04 | 0.00 | - | - |
| place wine | | 0.00 | 0.00 | 0.00 | 0.00 | 0.00 | 0.00 | 0.00 | - | - |
| Mean S.R. ↑ | | 0.32 | 0.23 | 0.34 | 0.15 | 0.18 | 0.37 | **0.39** | - | - |
| Mean Rank ↓ | | 3.05 | 4.35 | 3.15 | 4.75 | 4.70 | 2.65 | **2.15** | - | - |

| Tasks | | Diffusion Policy | | | | | | | | |
| --- | --- | --- | --- | --- | --- | --- | --- | --- | --- | --- |
| | | RGB | | RGB-D | | | Point Cloud | | Depth Only | |
| | | ● ResNet | ● ViT | ● ResNet | ● ViT | ● MultiViT | ● SpUNet | ● PointNet | ● ResNet | ● ViT |
| *ManiSkill2* | | | | | | | | | | |
| PickCube | | 0.17 | 0.24 | 0.34 | 0.58 | 0.00 | **0.71** | 0.70 | 0.04 | 0.01 |
| StackCube | | 0.03 | 0.00 | 0.59 | 0.03 | 0.00 | **0.04** | 0.00 | 0.00 | 0.00 |
| TurnFaucet | | 0.08 | 0.07 | 0.24 | 0.30 | 0.00 | 0.32 | **0.36** | 0.28 | 0.00 |
| Peg- | Grasp | 0.78 | 0.45 | 0.94 | 0.68 | 0.46 | 0.82 | **0.83** | 0.06 | 0.02 |
| Insertion- | Align | 0.07 | 0.02 | 0.11 | 0.03 | 0.02 | 0.09 | **0.16** | 0.00 | 0.00 |
| Side | Insert | **0.01** | 0.00 | 0.01 | 0.00 | 0.00 | **0.01** | 0.01 | 0.00 | 0.00 |
| Excavate | | 0.01 | 0.02 | 0.23 | 0.03 | 0.00 | 0.17 | **0.24** | 0.02 | 0.00 |
| Hang | | 0.52 | 0.42 | 0.77 | 0.56 | 0.00 | 0.67 | **0.72** | 0.65 | 0.09 |
| Pour | | 0.00 | 0.00 | 0.06 | 0.00 | 0.00 | 0.00 | 0.00 | 0.00 | 0.00 |
| Fill | | 0.36 | 0.04 | 0.72 | 0.03 | 0.01 | 0.21 | **0.68** | 0.21 | 0.02 |
| *RLBench* | | | | | | | | | | |
| open drawer | | 0.00 | 0.00 | 0.12 | 0.00 | 0.08 | **0.28** | 0.12 | - | - |
| sweep to | | 0.00 | 0.04 | 0.00 | 0.00 | 0.04 | 0.08 | **0.16** | - | - |
| meat off grill | | 0.00 | 0.00 | 0.00 | 0.00 | **0.12** | 0.00 | 0.00 | - | - |
| turn tap | | **0.24** | 0.04 | 0.04 | 0.12 | 0.16 | 0.16 | 0.16 | - | - |
| reach and drag | | **0.08** | 0.04 | 0.04 | 0.00 | 0.00 | 0.04 | **0.08** | - | - |
| put money | | 0.08 | 0.08 | 0.16 | 0.08 | **0.20** | 0.16 | 0.08 | - | - |
| push buttons | | 0.04 | 0.00 | 0.04 | 0.00 | **0.12** | 0.04 | **0.12** | - | - |
| close jar | | 0.00 | 0.00 | 0.00 | 0.00 | 0.00 | 0.00 | 0.00 | - | - |
| place wine | | **0.04** | 0.00 | 0.00 | 0.00 | 0.00 | 0.00 | 0.00 | - | - |
| Mean S.R. ↑ | | 0.13 | 0.08 | 0.23 | 0.13 | 0.06 | 0.20 | **0.23** | - | - |
| Mean Rank ↓ | | 3.37 | 4.47 | 2.37 | 4.00 | 4.21 | 2.42 | **1.89** | - | - |

tasks and has a lower mean rank, indicating instability. The primary issue lies in that depth data can exhibit high variability due to changes in object distance, leading to a more unstable data distribution. In robotic applications, the depth values of larger background areas often differ significantly from those of smaller foreground objects, complicating the learning process. These findings underscore the critical importance of explicit 3D representations, such as point clouds.

*Finding 3:* Using PVRs can lead to better performance *on average*, though not for all individual tasks. Although using point cloud has a higher baseline, ○ PonderV2 achieves a more significant performance gain compared to ○ R3M and is comparable to ○ VC-1 and ○ MultiMAE. Additionally, the size of ○ PonderV2's pre-training data is much smaller than that of the other PVRs—by orders of magnitude, from thousands (K) to millions (M). Unlike the other PVRs, ○ PonderV2 does not utilize object-centric or human interaction data, which are more aligned with robot domains. The reason may be attributed to **the multi-view rendering pretext task during ○ PonderV2's pre-training, which enriches it with more geometric knowledge, crucial for robot tasks.** This unexpected outcome suggests that, despite having less available data, *point cloud PVRs can still be highly efficient, and in some cases, even superior*.

Table 3: **Results of PVRs on different observation spaces.** Blue and red numbers denote the relative performance changes compared to their *corresponding training-from-scratch encoders.*

| Tasks | | ACT Policy | | | |
|---|---|---|---|---|---|
| | | ○ R3M | ○ VC-1 | ○ MultiMAE | ○ PonderV2 |
| *ManiSkill2* | | | | | |
| PickCube | | $\underline{0.82}^{\uparrow 0.22}$ | $0.77^{\uparrow 0.63}$ | $0.52^{\uparrow 0.49}$ | $\mathbf{0.87}^{\uparrow 0.13}$ |
| StackCube | | $\mathbf{0.41}^{\uparrow 0.09}$ | $0.06^{\uparrow 0.06}$ | $0.30^{\uparrow 0.30}$ | $\underline{0.35}^{\uparrow 0.13}$ |
| TurnFaucet | | $\mathbf{0.46}^{\downarrow 0.03}$ | $\underline{0.42}^{\uparrow 0.15}$ | $0.37^{\uparrow 0.02}$ | $0.27^{\downarrow 0.12}$ |
| Peg- | Grasp | $\mathbf{0.84}^{\uparrow 0.11}$ | $0.63^{\uparrow 0.27}$ | $\underline{0.71}^{\uparrow 0.55}$ | $0.65^{\downarrow 0.17}$ |
| Insertion- | Align | $\underline{0.23}^{\uparrow 0.06}$ | $0.07^{\uparrow 0.05}$ | $0.15^{\uparrow 0.14}$ | $\mathbf{0.23}^{\downarrow 0.05}$ |
| Side | Insert | $0.01^{\uparrow 0.01}$ | $0.00^{\uparrow 0.00}$ | $\underline{0.01}^{\uparrow 0.01}$ | $\mathbf{0.02}^{\uparrow 0.01}$ |
| Excavate | | $\underline{0.38}^{\uparrow 0.36}$ | $0.20^{\uparrow 0.20}$ | $0.28^{\uparrow 0.28}$ | $\mathbf{0.38}^{\uparrow 0.27}$ |
| Hang | | $\underline{0.84}^{\downarrow 0.02}$ | $\mathbf{0.84}^{\uparrow 0.04}$ | $0.77^{\downarrow 0.01}$ | $0.83^{\uparrow 0.04}$ |
| Pour | | $\mathbf{0.12}^{\uparrow 0.06}$ | $0.04^{\uparrow 0.04}$ | $0.00^{\uparrow 0.00}$ | $\underline{0.11}^{\uparrow 0.02}$ |
| Fill | | $\mathbf{0.88}^{\uparrow 0.09}$ | $\underline{0.78}^{\uparrow 0.48}$ | $0.68^{\downarrow 0.08}$ | $0.73^{\uparrow 0.17}$ |
| *RLBench* | | | | | |
| open drawer | | $0.00^{\uparrow 0.00}$ | $0.24^{\uparrow 0.08}$ | $\underline{0.36}^{\uparrow 0.16}$ | $\mathbf{0.60}^{\uparrow 0.16}$ |
| sweep to | | $0.52^{\downarrow 0.20}$ | $0.96^{\uparrow 0.16}$ | $\mathbf{1.00}^{\uparrow 0.32}$ | $\underline{0.96}^{\uparrow 0.06}$ |
| meat off grill | | $0.04^{\downarrow 0.20}$ | $\underline{0.12}^{\uparrow 0.04}$ | $0.04^{\uparrow 0.04}$ | $\mathbf{0.72}^{\uparrow 0.00}$ |
| turn tap | | $0.00^{\uparrow 0.00}$ | $\underline{0.04}^{\uparrow 0.04}$ | $\mathbf{0.08}^{\uparrow 0.08}$ | $0.00^{\uparrow 0.00}$ |
| reach and drag | | $0.04^{\downarrow 0.28}$ | $\underline{0.20}^{\downarrow 0.08}$ | $0.16^{\uparrow 0.12}$ | $\mathbf{0.28}^{\uparrow 0.08}$ |
| put money | | $0.48^{\downarrow 0.12}$ | $\underline{0.72}^{\downarrow 0.04}$ | $0.56^{\uparrow 0.28}$ | $\mathbf{0.64}^{\uparrow 0.04}$ |
| push buttons | | $0.24^{\uparrow 0.12}$ | $\underline{0.44}^{\uparrow 0.04}$ | $\mathbf{0.48}^{\uparrow 0.36}$ | $0.16^{\uparrow 0.12}$ |
| close jar | | $\underline{0.08}^{\uparrow 0.04}$ | $0.08^{\uparrow 0.08}$ | $0.00^{\uparrow 0.00}$ | $0.28^{\uparrow 0.24}$ |
| place wine | | $0.00^{\uparrow 0.00}$ | $0.00^{\uparrow 0.00}$ | $\underline{0.08}^{\uparrow 0.08}$ | $0.16^{\uparrow 0.16}$ |
| Mean S.R. ↑ | | $0.32^{\uparrow 0.00}$ | $\underline{0.34}^{\uparrow 0.10}$ | $0.33^{\uparrow 0.15}$ | $\mathbf{0.43}^{\uparrow 0.06}$ |
| Mean Rank ↓ | | $\underline{2.37}^{\uparrow 0.63}$ | $2.63^{\uparrow 0.16}$ | $2.68^{\downarrow 0.21}$ | $\mathbf{1.89}^{\uparrow 0.32}$ |

Table 4: **Mean S.R. on zero-shot camera view changes.**

| Methods | Vertical | | Horizontal | | Average |
|---|---|---|---|---|---|
| | 5° | 10° | 5° | 10° | |
| ● ResNet | 0.17 | 0.13 | 0.25 | 0.17 | 0.18 |
| ● ViT-B | 0.06 | 0.04 | 0.13 | 0.11 | 0.09 |
| ● MultiViT | 0.11 | 0.09 | 0.10 | 0.09 | 0.10 |
| ● SpUNet | 0.24 | 0.18 | 0.25 | 0.17 | 0.21 |
| ● PointNet | **0.39** | **0.38** | **0.43** | **0.41** | **0.40** |
| ○ R3M | 0.19 | 0.16 | 0.18 | 0.13 | 0.16 |
| ○ VC-1 | 0.13 | 0.11 | 0.16 | 0.17 | 0.14 |
| ○ MultiMAE | 0.21 | 0.15 | 0.23 | 0.18 | 0.19 |
| ○ PonderV2 | **0.29** | **0.21** | **0.29** | **0.22** | **0.25** |

Table 5: **Mean S.R. on sample efficiency.**

| #Training | 10 | 25 | 100 |
|---|---|---|---|
| ● ResNet | **0.009** | 0.009 | 0.227 |
| ● ViT-B | 0.000 | **0.040** | 0.284 |
| ● MultiViT | 0.004 | 0.027 | 0.149 |
| ● SpUNet | 0.004 | 0.018 | 0.322 |
| ● PointNet | 0.004 | 0.009 | **0.324** |
| ○ R3M | 0.009 | **0.067** | 0.317 |
| ○ VC-1 | 0.000 | 0.040 | 0.348 |
| ○ MultiMAE | 0.000 | 0.031 | 0.334 |
| ○ PonderV2 | **0.013** | 0.040 | **0.439** |

## 4.3 Study on Zero-Shot Generalization Capabilities Across Observation Spaces (Q3)

A key objective in robot learning is to build an agent with strong zero-shot generalization abilities. This section extensively evaluates the zero-shot generalization capabilities of different observation spaces, focusing on camera view (Sec. 4.3.1) and visual changes (Sec. 4.3.2). Note that the **RGB color is included in the feature of point clouds**, making the visual robustness non-trivial.

### 4.3.1 Zero-Shot Generalization to Camera View Changes

Camera view generalization is crucial for robot learning models but is often overlooked. Our study emphasizes its importance, conducting zero-shot evaluations of testing novel viewpoints by shifting

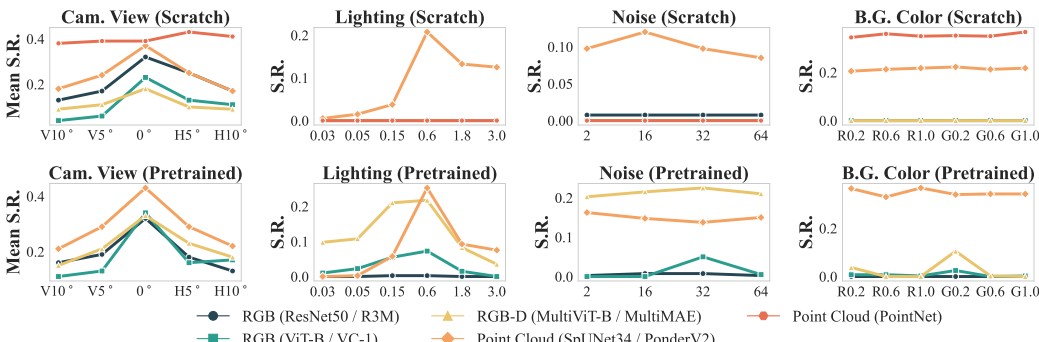

Figure 2: **Point cloud has better zero-shot generalization ability on camera view and visual changes.** We demonstrate the zero-shot generalization ability of different observation spaces. Encoders trained from scratch are shown in the first row, while PVRs are shown in the second row.

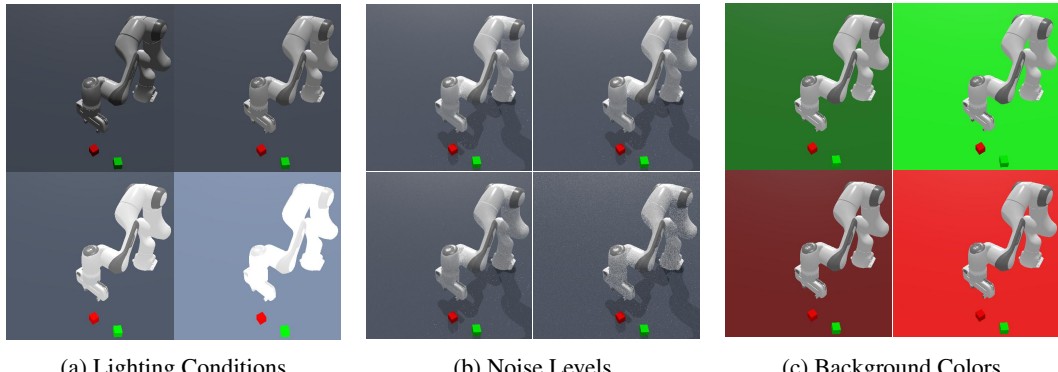

| (a) Lighting Conditions | (b) Noise Levels | (c) Background Colors |

Figure 3: **Examples of different visual changes.** (a) The light intensities are $0.03, 0.6, 0.15, and 0.3$ from left to right and from top to down respectively. (b) The ray tracing samples per pixel are $64, 32, 16, and 2$ respectively. (c) The background color is denoted as 'G0.2', 'G1.0', 'R0.2', and 'R1.0' respectively, where 'G' and 'R' means green or red and the number represents the value of the green or red channel.

the camera vertically and horizontally by $5°$ and $10°$ respectively. The mean success rates under these varied camera views are presented in Tab. 4 and visually represented in Fig. 2.

*Finding 4:* All methods are significantly affected by camera view changes, even with only 5 degrees. However, **point cloud data, both pre-trained and from scratch, shows notable resilience**. This suggests that image-based models are overly dependent on specific training views. Algorithms like [87, 9] use identical training and testing camera setups, leaving true robustness untested. Inferring 3D actions from images without camera parameters is inherently ill-posed. Our findings highlight the potential of point cloud representations for more robust robot models.

### 4.3.2 Zero-Shot Generalization to Visual Changes

We further discuss the problem of visual generalization. The `StackCube` task in ManiSkill2 was selected for its complexity and apparent preference for the RGB modality, as demonstrated by its higher initial accuracy. We consider 3 kinds of visual changes. **1) Lighting:** We vary lighting intensity, testing levels at $0.03, 0.05, 0.15, 0.6, 1.8, 3$, from the default intensity of $0.3$. **2) Noise:** We introduce visual noise by switching the rendering mode from rasterization to ray tracing, disabling the denoiser, and varying the number of ray tracing samples per pixel to $2, 16, 32$, and $64$. **3) Background color:** We alter the original gray floor to red and green, varying the 'R' or 'G' values to $0.2, 0.6, 1.0$ respectively. Illustrations of these visual changes are displayed in Fig. 3a, Fig. 3b and Fig. 3c. The results are plotted in Fig. 2. For comprehensive results on all our generalization experiments, please refer to Tab. 12, Tab. 13, Tab. 14, Tab. 15 and Tab. 16 in Appendix E.

*Finding 5:* We observe that *point cloud methods generally exhibit better generalization than other observation spaces*. Specifically, we find that • SpUNet demonstrates significantly greater robustness to foreground visual changes, whereas • PointNet's performance drops dramatically to near zero in these scenarios. Conversely, • PointNet shows superior generalization to changes in camera view. Given that point cloud inputs include both coordinate and color information, *these findings are quite noteworthy*. Sparse convolution operations maintain locality, which may contribute to robustness against noise. In contrast, point-based networks emphasize global information, which could enhance robustness to geometric changes.

*Finding 6:* Additionally, utilizing PVRs generally improves model generalization, **especially when semantic information is incorporated during pre-training**, as seen with ○ MultiMAE and ○ PonderV2. This semantic knowledge provides external invariance, enhancing model robustness.

### 4.4 Study on sample efficiency (Q4)

To evaluate the sample efficiency across different observation spaces, we conducted experiments on RLBench tasks with reduced training data. Specifically, each method was trained using only 10 and 25 out of the total 100 training trajectories. The results are presented in Tab. 5 and Appendix F.

Table 6: **Influence of different design choices on point cloud observations.** Results are reported for ManiSkill2 tasks. 'Pre. Samp.' denotes pre-sampling (FPS sampling before the encoder). 'Post. Samp.' indicates post-sampling (sampling after the encoder). Blue numbers show point cloud performance drop *relative to post-sampling with both color and coordinate information* (underlined). Red numbers indicate pointmap performance gain *compared to RGB image methods.* **Here, the novel pointmap format stacks both color and coordinate information within images.**

| Input | | ACT Policy | | | | | Diffusion Policy | | | |
|---|---|---|---|---|---|---|---|---|---|---|
| | Samp. | Encoder | Color | Coord. | Mean S.R. | Samp. | Encoder | Color | Coord. | Mean S.R. |
| Point Cloud | Pre. | ● SpUNet | ✓ | ✗ | $0.22^{\downarrow 0.18}$ | Pre. | ● SpUNet | ✓ | ✗ | $0.20^{\downarrow 0.21}$ |
| | | | ✗ | ✓ | $0.25^{\downarrow 0.15}$ | | | ✗ | ✓ | $0.15^{\downarrow 0.26}$ |
| | | | ✓ | ✓ | $0.21^{\downarrow 0.20}$ | | | ✓ | ✓ | $0.19^{\downarrow 0.22}$ |
| | | ● PointNet | ✓ | ✗ | $0.15^{\downarrow 0.29}$ | | ● PointNet | ✓ | ✗ | $0.16^{\downarrow 0.29}$ |
| | | | ✗ | ✓ | $0.29^{\downarrow 0.15}$ | | | ✗ | ✓ | $0.29^{\downarrow 0.17}$ |
| | | | ✓ | ✓ | $0.31^{\downarrow 0.13}$ | | | ✓ | ✓ | $0.39^{\downarrow 0.07}$ |
| | Post. | ● SpUNet | ✓ | ✗ | $0.23^{\downarrow 0.18}$ | Post. | ● SpUNet | ✓ | ✗ | $0.26^{\downarrow 0.15}$ |
| | | | ✗ | ✓ | $0.27^{\downarrow 0.14}$ | | | ✗ | ✓ | $0.30^{\downarrow 0.11}$ |
| | | | ✓ | ✓ | 0.41 | | | ✓ | ✓ | 0.41 |
| | | ● PointNet | ✓ | ✗ | $0.22^{\downarrow 0.23}$ | | ● PointNet | ✓ | ✗ | $0.18^{\downarrow 0.28}$ |
| | | | ✗ | ✓ | $0.38^{\downarrow 0.07}$ | | | ✗ | ✓ | $0.37^{\downarrow 0.08}$ |
| | | | ✓ | ✓ | **0.45** | | | ✓ | ✓ | **0.45** |
| Pointmap | N/A | ● ResNet | ✓ | ✓ | $0.43^{\uparrow 0.02}$ | N/A | ● ResNet | ✓ | ✓ | $0.28^{\uparrow 0.08}$ |
| | | ● ViT | ✓ | ✓ | $0.28^{\uparrow 0.09}$ | | ● ViT | ✓ | ✓ | $0.17^{\uparrow 0.04}$ |
| RGB-D | N/A | ● ResNet | ✓ | Depth | 0.34 | N/A | ● ResNet | ✓ | Depth | 0.03 |
| | | ● ViT | ✓ | Depth | 0.15 | | ● ViT | ✓ | Depth | 0.05 |

*Finding 7:* Our analysis reveals that point cloud observation spaces do not demonstrate a significant advantage in sample efficiency compared to other modalities. Notably, our results indicate that *PVRs consistently improve performance in scenarios with limited training data.* Remarkably, ○ PonderV2, despite having significantly less pre-training data, still shows notable enhancement. This suggests that leveraging pre-trained models can be particularly beneficial in few-shot learning contexts, where extensive training datasets are not available.

## 4.5 Study on design decisions on point cloud observation space (Q5)

In this section, we demonstrate that not only the point cloud itself matters but also how it is utilized is equally, if not more, important. Through extensive experimentation on ManiSkill2 tasks, we aim to provide insights into the use of point clouds in robotic learning problems. First, we investigate the importance of coordinate and color information. Next, we examine the influence of point cloud sampling strategies, particularly the widely adopted FPS sampling, used to obtain a fixed number of points for convenient input to policies, as seen in [73, 83]. Additionally, we explore the use of pointmap, a novel format that stacks RGB images with explicit coordinate information. The results of these experiments are detailed in Tab. 6. Full results of each task can be found in Appendix G.

*Finding 8:* **Post-sampling**, *i.e.*, FPS sampling on the feature map after the encoder, **can significantly enhance the performance of point cloud-based methods**, since it can maintain better *local information*. This finding is notable since most previous literature defaults to pre-sampling [73, 83].

*Finding 9:* Coordinate information is more critical than color information, as removing coordinate features results in a larger performance drop. Compared to the depth-only experiment results in Tab. 2, this further proves the necessity of utilizing explicit 3D structures. Moreover, using **both color and coordinate information yields the best results**.

*Finding 10:* Pointmap, which seemingly integrates the advantages of both 2D images and 3D information, consistently outperforms RGB-only and RGB-D methods. However, it still lags behind point clouds, especially when using diffusion policies. We believe this discrepancy arises because the

Table 7: Different observation spaces on the same encoder architectures with ACT policy.

| Tasks | | PointNet RGB | RGB-D | PCD | SpUNet RGB | RGB-D | PCD |
|---|---|---|---|---|---|---|---|
| PickCube | | 0.680 | 0.195 | **0.843** | 0.352 | 0.282 | **0.740** |
| StackCube | | 0.252 | 0.093 | **0.348** | 0.068 | 0.052 | **0.220** |
| TurnFaucet | | **0.393** | 0.335 | 0.000 | **0.447** | 0.268 | 0.388 |
| Peg- | Grasp | 0.675 | 0.565 | **0.765** | 0.658 | 0.668 | **0.810** |
| Insertion- | Align | 0.308 | 0.095 | **0.395** | 0.220 | 0.132 | **0.280** |
| Side | Insert | **0.027** | 0.000 | 0.005 | 0.007 | **0.013** | 0.008 |
| Excavate | | 0.035 | 0.038 | **0.268** | 0.058 | **0.097** | 0.032 |
| Hang | | 0.820 | **0.853** | 0.828 | 0.728 | 0.817 | **0.840** |
| Pour | | 0.112 | 0.110 | **0.135** | 0.090 | 0.000 | **0.095** |
| Fill | | 0.615 | 0.147 | **0.905** | **0.842** | 0.827 | 0.660 |
| Mean S.R. | | 0.392 | 0.243 | **0.449** | 0.347 | 0.316 | **0.407** |

Table 8: Different observation spaces on same encoder architectures with diffusion policy.

| Tasks | | PointNet RGB | RGB-D | PCD | SpUNet RGB | RGB-D | PCD |
|---|---|---|---|---|---|---|---|
| PickCube | | 0.863 | 0.430 | **0.900** | 0.623 | 0.567 | **0.710** |
| StackCube | | **0.495** | 0.140 | 0.238 | **0.143** | 0.120 | 0.035 |
| TurnFaucet | | 0.133 | 0.298 | **0.365** | 0.308 | 0.308 | **0.318** |
| Peg- | Grasp | 0.855 | 0.550 | **0.868** | 0.805 | 0.807 | **0.815** |
| Insertion- | Align | 0.243 | 0.072 | **0.285** | **0.143** | 0.072 | 0.093 |
| Side | Insert | **0.013** | 0.000 | 0.008 | 0.003 | **0.005** | 0.000 |
| Excavate | | 0.195 | 0.235 | **0.278** | 0.060 | 0.100 | **0.168** |
| Hang | | 0.740 | 0.712 | **0.778** | 0.540 | 0.600 | **0.673** |
| Pour | | 0.095 | 0.040 | **0.125** | **0.038** | 0.005 | 0.000 |
| Fill | | 0.252 | 0.165 | **0.693** | 0.203 | 0.195 | **0.208** |
| Mean S.R. | | 0.388 | 0.264 | **0.454** | 0.286 | 0.278 | **0.302** |

Table 9: Ablation on point cloud frames.

| Tasks | | ACT Policy PointNet World | EE | SpUNet World | EE | Diffusion Policy PointNet World | EE | SpUNet World | EE |
|---|---|---|---|---|---|---|---|---|---|
| PickCube | | 0.843 | **0.915** | **0.740** | 0.540 | 0.900 | **0.935** | 0.740 | **0.842** |
| StackCube | | 0.348 | **0.442** | 0.220 | **0.465** | **0.238** | 0.145 | 0.220 | **0.370** |
| TurnFaucet | | 0.000 | 0.000 | **0.388** | 0.000 | 0.365 | **0.545** | 0.388 | **0.435** |
| Peg- | Grasp | **0.765** | 0.647 | 0.810 | **0.873** | 0.868 | **0.890** | 0.810 | **0.910** |
| Insertion- | Align | **0.395** | 0.195 | **0.280** | 0.245 | **0.285** | 0.140 | **0.280** | 0.175 |
| Side | Insert | 0.005 | **0.007** | 0.008 | **0.015** | **0.008** | 0.001 | **0.008** | 0.000 |
| Excavate | | **0.268** | 0.005 | **0.032** | 0.025 | 0.278 | **0.278** | 0.113 | **0.245** |
| Hang | | **0.828** | 0.810 | **0.840** | 0.825 | 0.778 | **0.820** | **0.798** | 0.770 |
| Pour | | 0.135 | **0.172** | 0.095 | **0.096** | 0.125 | **0.140** | **0.095** | 0.000 |
| Fill | | 0.905 | **0.923** | **0.660** | 0.558 | 0.693 | **0.740** | **0.660** | 0.068 |

Table 10: Ablation on diffusion policy variants.

| Obs. Space | Encoder | Variant | Mean S.R. |
|---|---|---|---|
| RGB | ● ResNet50 | UNet | 0.165 |
| RGB-D | ● ResNet50 | UNet | 0.340 |
| Point Cloud | ● PointNet | UNet | 0.900 |
| Point Cloud | ● SpUNet | UNet | 0.740 |
| RGB | ● ResNet50 | Transformer | 0.280 |
| RGB-D | ● ResNet50 | Transformer | 0.340 |
| Point Cloud | ● PointNet | Transformer | 0.915 |
| Point Cloud | ● SpUNet | Transformer | 0.527 |

neighborhood locality in pointmaps is confined to 2D, whereas point clouds maintain 3D locality. This limitation may also be due to the insufficient research on encoders specifically tailored to the pointmap format. Nonetheless, the comparable results achieved using the ACT policy with point clouds suggest the significant potential value of further investigating this approach.

# 5   Additional Ablations

In this section, we further provide additional ablation experiments.

**Ablation study on encoder architecture.** We assigned XYZ coordinates to each image pixel based on its UV coordinates, treating images as "plain point clouds" for processing by point cloud networks. We experimented with both RGB and RGB-D modalities using ● PointNet and ● SpUNet, applying both ACT and diffusion policies. The results, shown in Tables 7 and 8, strongly support our claims.

**Ablation study on diffusion policy variants.** By default, we chose the UNet version of the diffusion policy for its stability and robustness. Many subsequent works, like [61, 83, 73], also use this variant. To validate this more clearly, we have conducted additional validations to compare UNet and transformer versions of diffusion policy under different observation spaces on the `PickCube` task. As shown in Tab. 10, the point cloud performs the best under different policy variants.

**Ablation study on point cloud coordinate frame.** In Frame Mining [49], the authors discovered that using the end-effector (EE) frame can lead to even better results. We conduct additional experiments with the EE frame using both ● PointNet and ● SpUNet. The results are in the Tab. 9. We found that the EE frame can indeed enhance point cloud performance in many cases, though this can vary depending on specific tasks and networks.

# 6   Real-World Experiments

As a benchmark work, our goal is to ensure our project is reproducible, open-source, easy to follow, and accessible to all researchers. In addition, some recent literature such as [46] indicates that the experimental conclusions in modern simulation environments are consistent with those in the real world. Thus we believe simulators can play a crucial role in providing fair and comparable evaluations. Besides, we also admit the importance of real-world validation. To address this, we conducted additional real-world experiments using the open-sourced low-cost-robot [43] equipped with two Intel

RealSense D415 RGB-D cameras. Our bimanual setups (including 2 leader arms, 2 follower arms, 2 cameras, etc.) cost about $2000 in total, making them affordable and easy to replicate for researchers.

We've also open-sourced our real-world codebase [13] for easy reproduction by other researchers. We designed three tasks: **1) Reach Cube:** A single arm with rigid objects. The robot is required to reach towards a cube and touch it. **2) Pick Cube:** A single arm with rigid objects. The robot is required to pick up the cube and hold it in the air. **3) Fold Cloth:** Two dual arms with soft-body objects. The robot is required to simultaneously catch one side of the

Table 11: Real-world results.

| Obs. Space | Reach Cube | Pick Cube | Fold Cloth |
|---|---|---|---|
| RGB | 0.60 | 0.05 | 0.65 |
| RGB-D | 0.30 | 0.20 | 0.50 |
| Point Cloud | **0.80** | **0.40** | **0.80** |

cloth with two grippers and then fold the cloth in half. The detailed setups, our workstation, and task visualizations are shown in Appendix D. The real-world results shown in Tab. 11 align with our simulated experiments, further supporting our conclusions.

## 7 Related Work

**Pre-trained Visual Representations (PVRs).** Recent advancements in self-supervised learning (SSL) for 2D and 3D computer vision, using methods like contrastive learning [7, 33, 8, 75, 35, 74, 63], distillation-based technique [6, 2, 55]s, reconstructive approaches [3, 34, 57, 85, 78], and differentiable rendering for pre-training [36, 90, 79] have shown promise. These techniques have been adapted for Embodied AI and robot learning, achieving notable results [58, 54, 64, 51, 42, 77, 52, 22, 80]. However, most focus on 2D RGB spaces, overlooking diverse observation spaces and 3D PVRs. Our research explores the impact of PVRs on different observation spaces.

**Observation Spaces in Robot Learning.** Estimating states from raw observations in robot learning has mainly focused on 2D RGB images. However, the importance of 3D information is gaining recognition [84, 27, 67, 68, 26, 81, 24, 47, 11, 82], as seen in methods like CLIPort [67] and Perceiver-Actor [68]. Despite progress, there is a lack of systematic comparison between 2D and 3D methods, with some 3D techniques being complex [26, 47, 11] or reliant on dense voxel representations [68, 82].

**Robot Learning for Manipulation.** Modeled as (Partially-Observable) Markov Decision Processes [41, 44], robot manipulation in RL faces issues like multi-objectiveness, non-stationarity, sim-to-real gaps, and poor sample efficiency [71, 25, 31, 65]. Behavior cloning [60, 45] offers a successful alternative with diverse approaches [68, 4, 5, 40, 87, 9], *etc*. Current foundational models [56, 72] primarily use RGB images for observation. Our research examines the impact of different observation spaces to guide future advancements.

## 8 Conclusion and Limitations

In this study, we introduce OBSBench to advance research on various observation spaces in robot learning. Our findings indicate that point cloud methods consistently outperform RGB and RGB-D in terms of success rate and robustness across different conditions, regardless of whether they are trained from scratch or pre-trained. Design choices, such as post-sampling and the inclusion of coordinate and color information, further enhance their performance. However, point cloud methods face challenges with sample efficiency. Utilizing large-scale 3D datasets like RH20T [23] and DL3DV-10K [48] could improve their robustness and generalization. Future research should explore dynamic sampling techniques and multi-modal integration, including tactile sensing. Although our experiments are conducted on simulated benchmarks to ensure consistency and fairness, translating and validating these findings in real-world scenarios in a *reproducible* and *credible* manner remains an open question. In the short term, we do not foresee any negative societal impacts from this work. However, as our results contribute to the development of more robust robotic systems, it is crucial to study how to prevent robots from causing harm in daily life in the long run.

## Acknowledgments and Disclosure of Funding

This work is funded in part by the National Key R&D Program of China No.2022ZD0160102, and Shanghai Artificial Intelligence Laboratory.

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

# A    Task Details and Examples

In our OBSBench, we support over 125 tasks in the ManiSkill2 and RLBench simulators. However, due to computational constraints, we selected 19 diverse and representative tasks for our experiments. To maintain generality, we chose tasks that 1) require distinct skills and 2) have non-trivial performance outcomes. For example, we did not select `PickYCBObject` in ManiSkill2 as it involves the same skill as `PickCube-v0`. Similarly, we excluded `stack blocks` in RLBench since all observation spaces resulted in a 0 success rate, indicating the bottleneck lies in the policy rather than the observation spaces. Below, we provide details and examples of our selected tasks.

## A.1    Maniskill2

### A.1.1    PickCube-v0

- **Objective:** Pick up a cube and move it to a goal position.
- **Success Criteria:** The cube is within 2.5 cm of the goal position, and the robot is static.
- **Demonstration:** 1000 successful trajectories.
- **Oracle Trajectory:** Shown in Figure A.1.

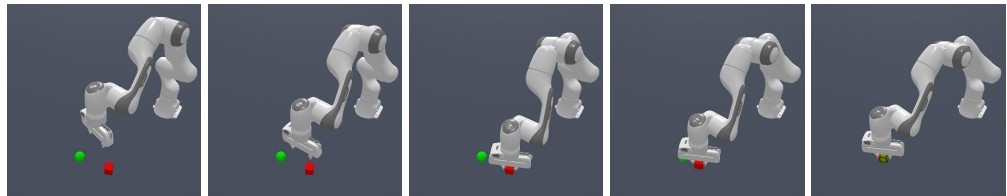

Figure A.1: Illustrations on ManiSkill2 task: `PickCube-v0`.

### A.1.2    StackCube-v0

- **Objective:** Pick up a red cube and place it onto a green one.
- **Success Criteria:** The red cube is placed on top of the green one stably and it is not grasped.
- **Demonstration:** 1000 successful trajectories.
- **Oracle Trajectory:** Shown in Figure A.2.

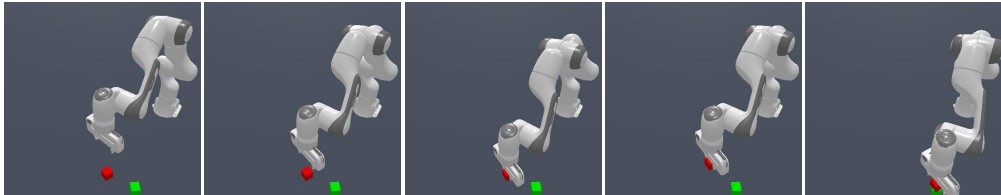

Figure A.2: Illustrations on ManiSkill2 task: `StackCube-v0`.

### A.1.3    TurnFaucet-v0

- **Objective:** Turn on a faucet by rotating its handle.
- **Success Criteria:** The faucet handle has been turned past a target angular distance.
- **Demonstration:** From the original set of 5510 trajectories (100 trajectories per faucet for most of 60 models from PartNet-Mobility), our experiment focused on 10 models, totaling 1000 trajectories.
- **Oracle Trajectory:** Shown in Figure A.3.

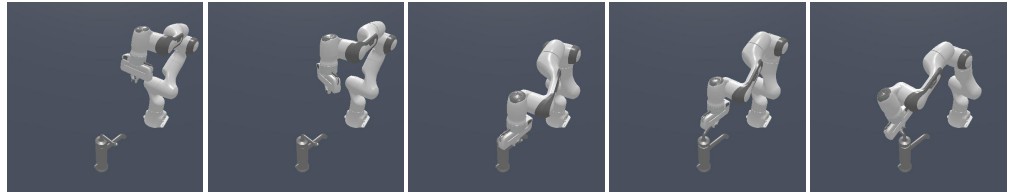
Figure A.3: Illustrations on ManiSkill2 task: `TurnFaucet-v0`.

### A.1.4 PegInsertionSide-v0

- **Objective:** Pick up the peg, align it with the horizontal hole in the box, and then insert the peg into the hole.
- **Success Criteria:** Following [39], the task is divided into three subtasks. The success criterion for the first subtask is picking up the peg. In the second subtask, success is achieved by aligning the peg such that both its head and entirety are within 1 cm of the target hole on the YZ plane. The final subtask is completed when half of the peg is inserted into the hole.
- **Demonstration:** 1000 successful trajectories.
- **Oracle Trajectory:** Shown in Figure A.4.

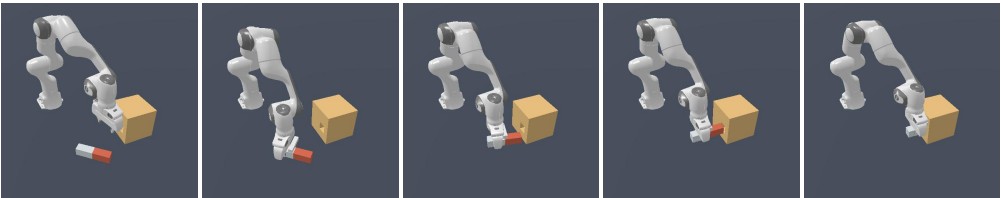
Figure A.4: Illustrations on ManiSkill2 task: `PegInsertionSide-v0`.

### A.1.5 Excavate-v0

- **Objective:** Lift a specific amount of clay to a target height.
- **Success Criteria:** The amount of lifted clay must be within a given range; the lifted clay is higher than a specific height; fewer than 20 clay particles are spilled on the ground; soft body velocity<0.05.
- **Demonstration:** 200 successful trajectories.
- **Oracle Trajectory:** Shown in Figure A.5.


Figure A.5: Illustrations on ManiSkill2 task: `Excavate-v0`.

### A.1.6 Hang-v0

- **Objective:** Hang a noodle on a target rod.
- **Success Criteria:** Part of the noodle is higher than the rod; two ends of the noodle are on different sides of the rod; the noodle is not touching the ground; the gripper is open; soft body velocity<0.05.
- **Demonstration:** 200 successful trajectories.
- **Oracle Trajectory:** Shown in Figure A.6.

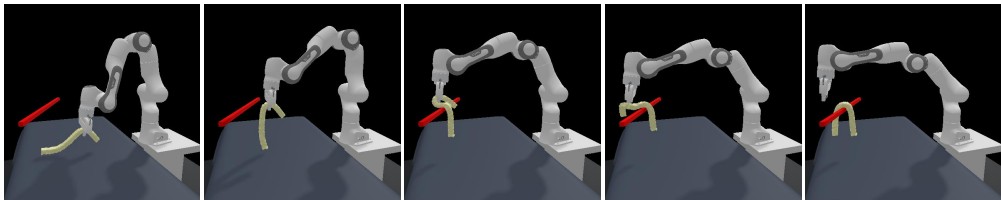
Figure A.6: Illustrations on ManiSkill2 task: `Hang-v0`.

### A.1.7 Pour-v0

- **Objective:** Pour liquid from a bottle into a beaker.
- **Success Criteria:** The liquid level in the beaker is within 4mm of the red line; the spilled water is fewer than 100 particles; the bottle returns to the upright position in the end; robot arm velocity<0.05.
- **Demonstration:** 200 successful trajectories.
- **Oracle Trajectory:** Shown in Figure A.7.

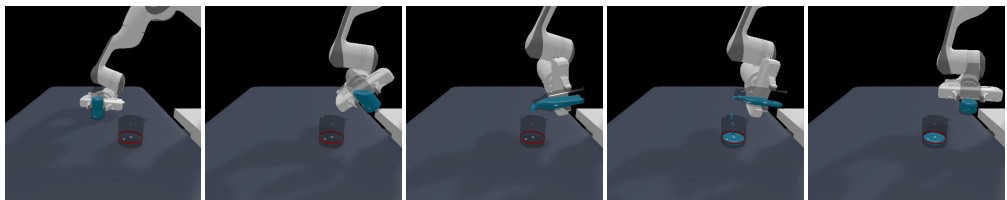
Figure A.7: Illustrations on ManiSkill2 task: `Pour-v0`.

### A.1.8 Fill-v0

- **Objective:** Fill clay from a bucket into the target beaker.
- **Success Criteria:** The amount of clay inside the target beaker>90%; soft body velocity<0.05.
- **Demonstration:** 200 successful trajectories.
- **Oracle Trajectory:** Shown in Figure A.8.

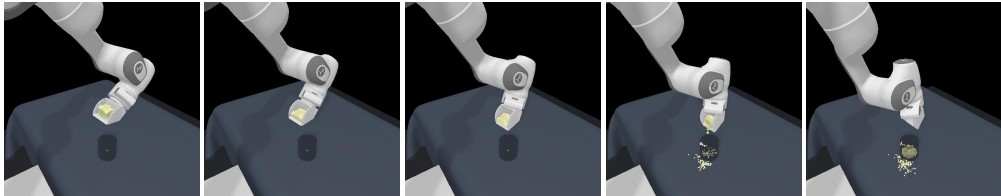
Figure A.8: Illustrations on ManiSkill2 task: `Fill-v0`.

## A.2 RLBench

All RLBench tasks are trained on 100 successful trajectories. Each task has multiple variations with different language descriptions.

### A.2.1 Open Drawer

- **Objective:** Open one of the three drawers: top, middle, or bottom.
- **Success Criteria:** The prismatic joint of the specified drawer is fully extended.
- **Example description:** Open the top drawer.
- **Oracle Trajectory:** Shown in Figure A.9.

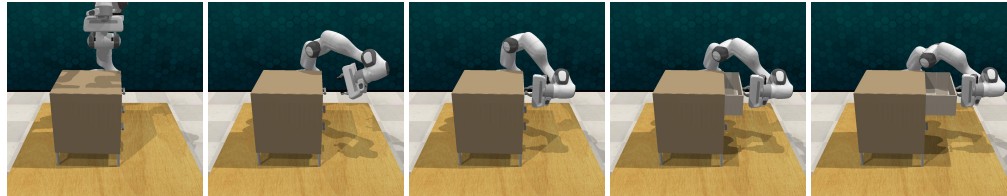
Figure A.9: Illustrations on RLBench task: open drawer.

### A.2.2 Sweep to Dustpan of Size

- **Objective:** Use the broom to brush the dirt particles into either the short or tall dustpan.
- **Success Criteria:** All 5 dirt particles are inside the specified dustpan.
- **Example description:** Sweep dirt to the short dustpan.
- **Oracle Trajectory:** Shown in Figure A.10.

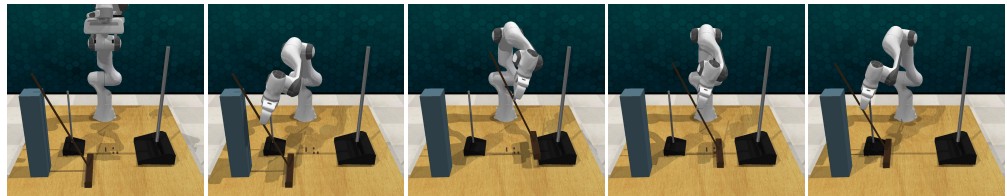
Figure A.10: Illustrations on RLBench task: sweep to dustpan of size.

### A.2.3 Meat off Grill

- **Objective:** Take either the chicken or steak off the grill and set it down on the side.
- **Success Criteria:** The specified meat is on the side, away from the grill.
- **Example description:** Take the steak off the grill.
- **Oracle Trajectory:** Shown in Figure A.11.

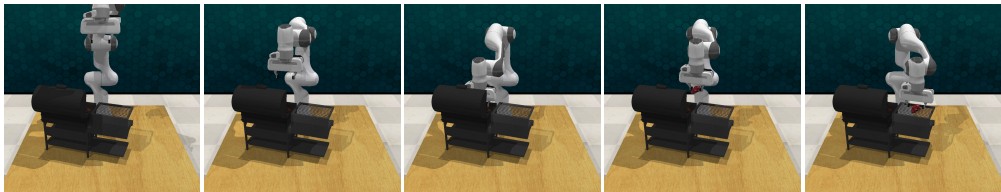
Figure A.11: Illustrations on RLBench task: meat off grill .

### A.2.4 Turn Tap

- **Objective:** Turn either the left or right handle of the tap. Left and right are defined with respect to the faucet orientation.
- **Success Criteria:** The revolute joint of the specified handle is at least $90°$ off from the starting position.
- **Example description:** Turn right tap.
- **Oracle Trajectory:** Shown in Figure A.12.

### A.2.5 Reach and Drag

- **Objective:** Grab the stick and use it to drag the cube onto the specified colored target square. The target colors are sampled from the full set of 20 color instances.

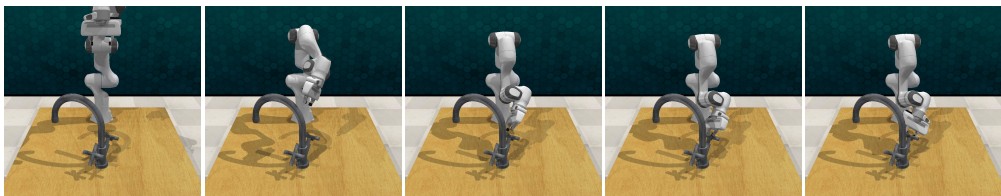
Figure A.12: Illustrations on RLBench task: `turn tap` .

- **Success Criteria:** Some part of the block is inside the specified target area.
- **Example description:** Use the stick to drag the cube onto the navy target.
- **Oracle Trajectory:** Shown in Figure A.13.

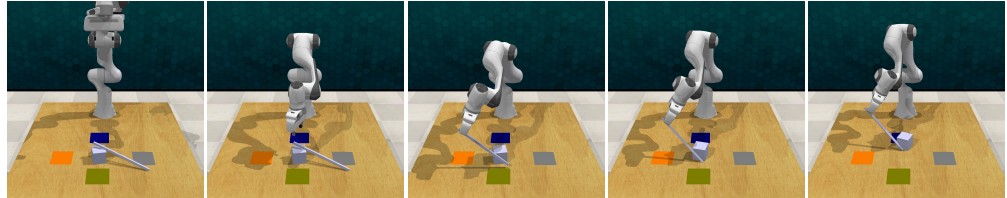
Figure A.13: Illustrations on RLBench task: `reach and drag` .

### A.2.6 Put Money in Safe

- **Objective:** Pick up the stack of money and put it inside the safe on the specified shelf. The shelf has three placement locations: top, middle, and bottom.
- **Success Criteria:** The stack of money is on the specified shelf inside the safe.
- **Example description:** Put the money away in the safe on the top shelf.
- **Oracle Trajectory:** Shown in Figure A.14.

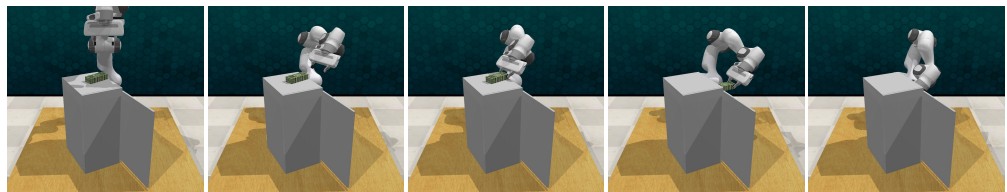
Figure A.14: Illustrations on RLBench task: `put money in safe`.

### A.2.7 Push Buttons

- **Objective:** Push the colored buttons in the specified sequence. The button colors are sampled from the full set of 20 color instances. There are always three buttons in the scene.
- **Success Criteria:** All the specified buttons were pressed.
- **Example description:**Push the maroon button, then push the green button, then push the navy button.
- **Oracle Trajectory:** Shown in Figure A.15.

### A.2.8 Close Jar

- **Objective:** Pick up the lid and close the jar of the specified color. The target colors are sampled from the full set of 20 color instances.
- **Success Criteria:**The lid is capped on a specific jar.
- **Example description:** Close the red jar.
- **Oracle Trajectory:** Shown in Figure A.16.

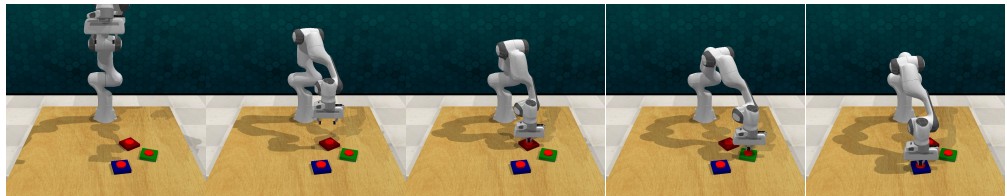
Figure A.15: Illustrations on RLBench task: push buttons.

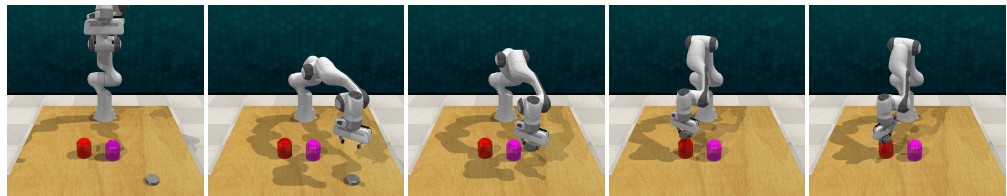
Figure A.16: Illustrations on RLBench task: close jar.

### A.2.9 Place Wine at Rack Location

- **Objective:** Grab the wine bottle and put it on the wooden rack at one of the three specified locations: left, middle, right. The locations are defined with respect to the orientation of the wooden rack.
- **Success Criteria:** The wine bottle is at the specified placement location on the wooden rack.
- **Example description:** Stack the wine bottle to the left of the rack.
- **Oracle Trajectory:** Shown in Figure A.17.

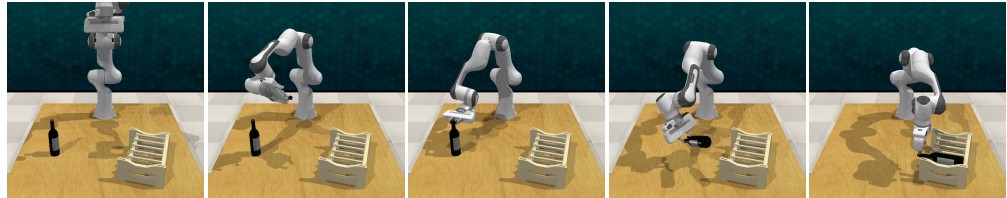
Figure A.17: Illustrations on RLBench task: place wine at rack location.

## B  Method Details

In this section, we outline our implementation details. For more information, we recommend readers refer to our GitHub repository: https://github.com/HaoyiZhu/PointCloudMatters.

### B.1  Encoders

• **ResNet50.** A key element in the • ResNet50 [32] architecture is its utilization of deep residual learning with skip connections, a design that effectively addresses the vanishing gradient problem in deep networks. This structure allows • ResNet50 to develop robust feature representations, crucial for complex image processing tasks.

• **ViT-B (Vision Transformer).** Representing a significant shift in image processing, • ViT-B [18] applies the transformer mechanism, traditionally used in natural language processing, to visual data. It processes images by segmenting them into patches and analyzing these via a transformer encoder, adept at capturing intricate spatial hierarchies.

• **MultiViT-B.** An innovative adaptation of the Vision Transformer, • MultiViT-B [1] is tailored for multi-modal input, including RGB-D images. It features unique projection layers for each modality,

seamlessly integrating depth information with RGB data. This fusion enriches the model's perception of spatial relationships, enhancing its environmental analysis capabilities.

• **SpUNet34.** In the realm of 3D vision, • SpUNet [14] is a prominent choice for processing sparse 3D data structures like point clouds. It employs sparse convolution, efficiently handling non-uniform data distribution in 3D space. The SpUNet architecture, inspired by ResNet34, is modified to accommodate the nuances of 3D point cloud data, ensuring effective processing and interpretation of these complex structures.

• **PointNet.** • PointNet [62] revolutionized the processing of point cloud data by directly consuming raw point sets without requiring voxelization or other preprocessing steps. It uses a series of multi-layer perceptrons (MLPs) to independently process each point, followed by a symmetric function to aggregate global features. This design allows • PointNet to handle unordered point sets efficiently, capturing both local and global structures. • PointNet's simplicity and effectiveness have made it a foundational model for various 3D deep learning tasks, including object classification, part segmentation, and scene segmentation.

## B.2  Pretrained Visual Representations

○ **R3M.** Employed for • ResNet50, ○ R3M [54] uses time-contrastive learning, video-language alignment, and L1 regularization to produce sparse, compact representations. It's trained on a substantial 3,500 hours of human interaction videos from the Ego4D [29] dataset, demonstrating superior performance over other methods like CLIP [63] and MoCo [33] in simulated robotic manipulation tasks.

○ **VC-1.** This approach pretrains a ViT using masked auto-encoding (MAE) [34], similar to MVP [64] but with a broader dataset range, including Ego4D [29], 100 Days of Hands (100DOH) [66], Something-Something v2 (SS-V2) [28], Epic Kitchens [16], Real Estate 10K [88], and ImageNet [17]. It excels in diverse embodied AI tasks.

○ **MultiMAE (Multi-modal Multi-task Masked Autoencoders).** ○ MultiMAE [1] is utilized for MultiViT, featuring masked autoencoding across various modalities including RGB, depth, and semantics on the ImageNet-1K dataset. Its cross-modality predictive coding significantly enhances transfer to downstream tasks such as image classification and depth estimation. In our study, we focus on its RGB and depth components.

○ **PonderV2.**  As the state-of-the-art self-supervised learning method for point clouds, ○ PonderV2 [90] employs differentiable neural rendering. The method trains a 3D backbone (• SpUNet34) within a volumetric neural renderer, focusing on learning detailed geometry and appearance cues. ○ PonderV2 is effective across a wide range of tasks, including high-level challenges like 3D detection and segmentation, as well as low-level objectives like 3D reconstruction and image synthesis, covering both indoor and outdoor scenarios. It demonstrates the framework's effectiveness across multiple benchmarks and showcases its potential in advancing 3D foundation models.

## B.3  Policy Network

In our study, we implemented the Action Chunking Transformer (ACT) [87] and Diffusion Policy [9] as our primary policy network. This choice was driven by their proficiency in addressing the inherent challenges of imitation learning, particularly in precision-demanding contexts where errors tend to accumulate progressively.

**Action Chunking Transformer (ACT)** stands out for its innovative approach to generative modeling over action sequences, significantly enhancing the robot's ability to execute complex tasks with heightened accuracy. A pivotal feature of ACT is its action chunking technique, which involves grouping sequences of actions into cohesive units for execution. This method effectively shortens the task's horizon and minimizes the accumulation of errors, making it particularly beneficial for tasks that require detailed, fine-grained manipulations. Furthermore, ACT integrates a conditional variational autoencoder (CVAE) architecture. This design excels at accommodating the variability and stochastic nature of human demonstrations, a common challenge in robot learning. The CVAE framework facilitates the efficient learning of nuanced manipulation skills, enabling ACT to surpass previous methodologies in both simulated and real-world applications.

**Diffusion Policy**, on the other hand, leverages the principles of diffusion processes to model the distribution over action sequences. This approach is particularly effective in high-dimensional action spaces where traditional methods struggle. The Diffusion Policy framework models the action distribution as a series of gradual transformations, allowing for more robust and flexible policy learning. One of the core advantages of the Diffusion Policy is its ability to handle the inherent uncertainty and variability in robot tasks by progressively refining actions through a diffusion process. This iterative refinement helps in reducing the noise and improving the precision of the actions, which is crucial for tasks requiring high accuracy and robustness.

## C   Implementation Details

**Experimental Settings.** For all experiments, we use a single front (base) camera to get input observations with a resolution of $128 \times 128$ for a simple and fair comparison. We evaluate each ManiSkill2 task using 400 fixed random seeds. In the RLBench experiments, we follow the settings of PerAct [37], training each task with 100 demonstrations and testing them on 25 distinct, held-out scenes. For all experiments, we do *not* use any data augmentation tricks for fairness and simplicity.

**Training Details.** For training each method, we employ the AdamW [50] optimizer, known for its efficiency in weight optimization, with a weight decay set at 0.05. We configure the OneCycle [69] learning rate scheduler to accelerate the learning process, with a pct_start of 0.15, an anneal_strategy set to 'cos', a div_factor at 100, and a final_div_factor at 1000. The scheduler is updated at a step interval frequency of 1. We conduct experiments across learning rates of $1e-5, 5e-5, 1e-4$, each replicated twice to ensure consistency. Performance evaluation relies on the last checkpoint or the one with the lowest validation loss. All experiments are conducted on a single NVIDIA RTX 3090 or 4090 GPU with a batch size of 32. For the ACT policy experiments, we train for 500 epochs, while the diffusion policy experiments are trained for 1800 epochs. For the RLBench experiments, we deviate from the quaternion representation and instead use a 6D rotation [89] representation for rotation actions, following [26]. In the ACT experiments, we apply action chunking and temporal ensembling with an exponential decay factor of $k = 0.01$, consistent with the original methodology.

**General Implementations.** Our implementation leverages PyTorch [59], a powerful framework for deep learning applications. For convenient and flexible experimental configuration, we build our codebase upon PyTorch Lightning [21] and Hydra [76]. For the ● ResNet50 encoder, we utilized the official model available in TorchVision [53], aligning with the implementation in ○ R3M. This approach ensures consistency with the original work regarding weights and feature extraction. We meticulously adhered to the original implementations for the ● ViT and ○ VC-1 models. In our experiments, we chose the ViT-base and corresponding VC-1-base models to maintain parameter consistency with other methods used in our study, promoting relative fairness. In the case of ● MultiViT and ○ MultiMAE, our implementation is based on the official ● MultiViT-B model. We employed the strongest pre-trained weights in the official repository, pre-trained simultaneously on RGB, depth, and semantic modalities, enhancing the model's robustness and versatility. For ● SpUNet and ○ PonderV2, we adopt the ● SpUNet34 architecture as described in the official ○ PonderV2 paper, ensuring alignment with their proposed methodology and optimizing the model's performance in handling point cloud data. The point cloud-related operations are largely adapted from Pointcept [12].

**Input Pre-processing.** For inputs to ● ResNet50 and ○ R3M, we rigorously adhere to the ○ R3M pre-processing guidelines. This includes resizing images to $224 \times 224$ and normalizing them using the ImageNet mean and standard deviation. For ● ViT and ○ VC-1, we follow the ○ VC-1 pre-processing protocol. This process involves resizing images to $256 \times 256$ via bicubic interpolation, center cropping them to $224 \times 224$ and then normalizing with ImageNet mean and std. For ● MultiViT and ○ MultiMAE inputs, we meticulously replicate the original method's pre-processing. This includes resizing, center cropping, and normalizing the RGB component, along with truncating and normalizing the depth images. For point cloud data, we employ grid sampling with a grid size of 0.005, aligning with the PonderV2 methodology. Notably, the original ○ PonderV2 model utilizes 6-channel inputs of RGB colors and normals. In our case, given the absence of normal values in robotic data, we substitute XYZ values alongside RGB. Despite this deviation from the original training distribution, we surprisingly find that this approach yields significant performance improvements after end-to-end finetuning. For the PointNet implementation, we follow the Minkowski Engine [10]'s approach to use sparse convolution. This is because each input batch has a different number of points, and the $1 \times 1$ convolution in PointNet is equivalent to a fully connected layer.

**Feature Extraction.** In line with the original ACT implementation, we extract features from the final convolutional layer of ResNet. To enhance these features, we also follow ACT to incorporate 2D cosine position embeddings, enriching the spatial context of the extracted feature map. For ViT and MultiViT, we focus on the `[CLS]` token, recognized for encapsulating global information of the input. We treat this token as a $1 \times 768$ feature map and apply a position embedding approach analogous to that used for ResNet's features. In processing point cloud data, our initial step involves utilizing Farthest Point Sampling (FPS) to select $2048$ seed points. For each seed point, we employ a K-Nearest Neighbors (KNN) algorithm to cluster $16$ neighboring points. Subsequently, each cluster undergoes max pooling following a linear projection layer. The final step entails adding positional embeddings based on the 3D coordinates of the seed points, which enhances the spatial relevance of the extracted features.

**ACT Policy Network.** In implementing the Action Chunking Transformer (ACT) policy network, we adhere to the original framework while tailoring the encoder forward function to suit our specific needs. Key hyper-parameters are chosen as follows: we set the dropout rate to $0.1$, the number of heads (head) to $8$, the dimension of the feedforward network (dim_feedforward) to $32$, the number of encoder layers to $4$, and decoder layers to $7$. The hidden dimension (hidden_dim) is fixed at $512$, with chunk sizes of $100$ for ManiSkill2 and $20$ for RLBench tasks. The latent dimension (latent_dim) is configured to $32$, and the weight for the KL divergence loss is set at $10.0$. For tasks involving goal conditions, we project the goals into a $512$-dimensional space to align with the hidden dimension of the transformer policy. In the case of language-based goals in RLBench, we employ the language encoder from CLIP to extract a feature vector of $512$ dimensions.

**Diffusion Policy Network.** We follow the official UNet version of the diffusion policy, as it is stable to hyper-parameters. The hyperparameters are set to be the same as the original implementation. Specifically, we use a horizon of $16$, number of action steps (n_action_steps) of $8$, and number of observation steps (n_obs_steps) of $2$. We set the obs_as_global_cond parameter to be true, the diffusion step embed dimension to be $128$, the down dimensions to be $[512, 1024, 2048]$, the kernel size to be $5$, the number of groups to be $8$, and the cond_predict_scale to be true.

## D    Real-World Setups

In real-world experiments, we implement three modalities using • ResNet50 encoders for RGB and RGB-D images, and • PointNet for point cloud inputs. The ACT policy is employed across all experiments. All modalities share the same settings and hyper-parameters. For data collection, leader arms are used to teleoperate the follower arms. Two RealSense RGB-D cameras capture the visual observations. Training is conducted on a single NVIDIA A100 GPU with a learning rate of $5e - 5$.

- **Reach Cube:** Trained with 45 demonstrations over 100 epochs.

- **Pick Cube:** Trained with 45 demonstrations over 500 epochs.

- **Fold Cloth:** Trained with 50 demonstrations over 1000 epochs.

All models are evaluated with 20 rollouts in the same environment. We report the success rates below, and corresponding videos are available at Google Drive (https://drive.google.com/drive/folders/1UiFgHv9QUPEM2is-N1OIJ47DjeYiiFEm?usp=drive_link). The real-world results from these tasks align with our simulated experiments, further supporting our conclusions.

Our workstation is illustrated in Fig. D.18 and our tasks are illustrated in Fig. D.19.

## E    Full Zero-Shot Generalization Results

Here we give out full results on zero-shot generalization experiments.

### E.1    Zero-Shot Generalization to Camera View

The results on camera view changes are listed in Tab. 12, Tab. 13, Tab. 14, Tab. 15.

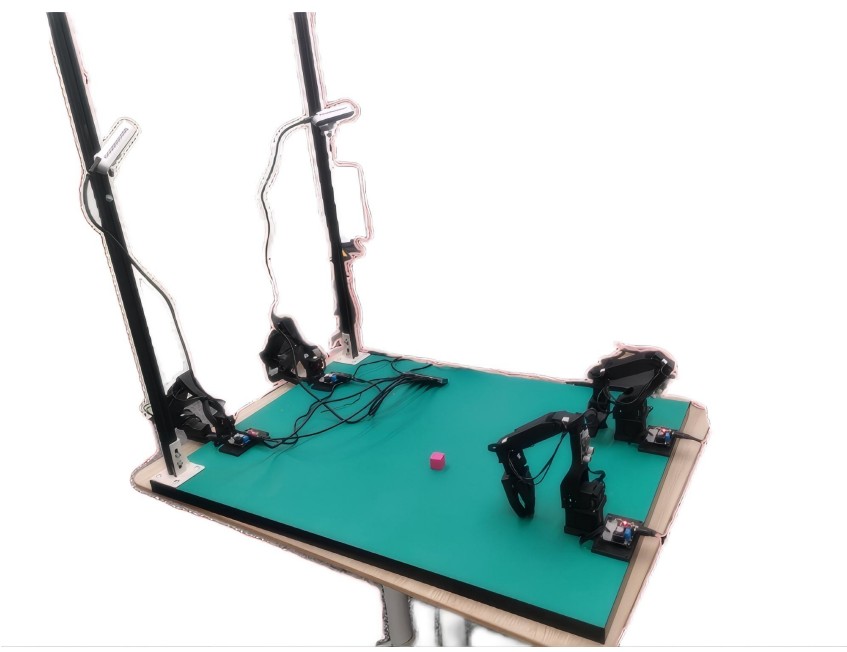

Figure D.18: **Our real-world workstation.**

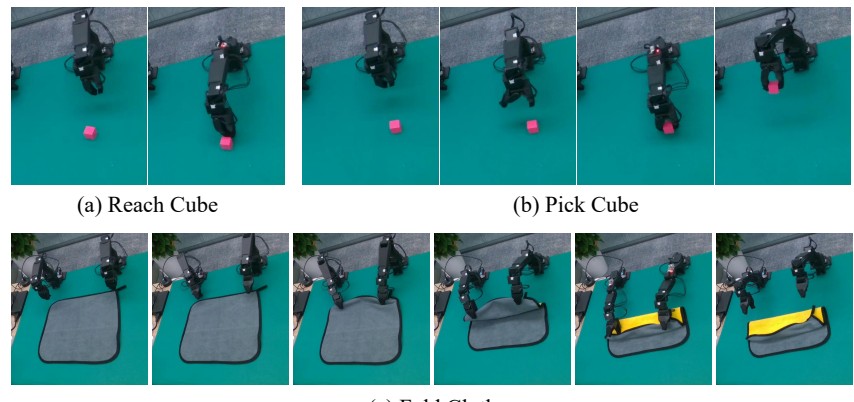

(a) Reach Cube                    (b) Pick Cube

(c) Fold Cloth

Figure D.19: **Our real-world tasks.**

## E.2    Zero-Shot Generalization to Visual Changes

The results of visual changes are listed in Tab. 16.

## F    Full Sample Efficiency Results

Full sample efficiency results on RLBench are listed in Tab. 17.

## G    Full Design Decision Results

Full design decision results on ManiSkill2 are listed in Tab. 18.

Table 12: Full results on the zero-shot generalization of scratch encoders to vertical camera view changes.

| Tasks | | Vertical 5° | | | | | Vertical 10° | | | | |
|---|---|---|---|---|---|---|---|---|---|---|---|
| | | ● ResNet | ● ViT | ● MultiViT | ● SpUNet | ● PointNet | ● ResNet | ● ViT | ● MultiViT | ● SpUNet | ● PointNet |
| *ManiSkill2* | | | | | | | | | | | |
| PickCube | | 0.66 | 0.13 | 0.02 | 0.64 | **0.83** | 0.58 | 0.03 | 0.02 | 0.44 | **0.85** |
| StackCube | | 0.32 | 0.00 | 0.00 | 0.21 | **0.36** | 0.26 | 0.00 | 0.00 | 0.22 | **0.37** |
| TurnFaucet | | 0.08 | 0.06 | **0.14** | 0.09 | 0.00 | 0.09 | 0.07 | **0.13** | 0.09 | 0.00 |
| Peg- | Grasp | 0.62 | 0.33 | 0.15 | **0.81** | 0.76 | 0.45 | 0.27 | 0.14 | **0.81** | 0.76 |
| Insertion- | Align | 0.09 | 0.02 | 0.00 | 0.30 | **0.39** | 0.04 | 0.02 | 0.00 | 0.28 | **0.40** |
| Side | Insert | 0.00 | 0.00 | 0.00 | 0.01 | **0.01** | 0.00 | 0.00 | 0.00 | 0.01 | **0.01** |
| Excavate | | 0.00 | 0.00 | 0.00 | 0.00 | **0.01** | 0.00 | 0.00 | 0.00 | 0.04 | **0.05** |
| Hang | | 0.75 | 0.51 | 0.81 | 0.62 | **0.96** | 0.61 | 0.33 | 0.67 | 0.71 | **0.95** |
| Pour | | 0.01 | 0.00 | 0.00 | 0.08 | **0.10** | 0.01 | 0.00 | 0.00 | 0.06 | **0.10** |
| Fill | | 0.29 | 0.05 | 0.75 | 0.53 | **0.92** | 0.34 | 0.10 | 0.76 | 0.15 | **0.81** |
| *RLBench* | | | | | | | | | | | |
| open drawer | | 0.00 | 0.00 | 0.00 | 0.00 | 0.00 | 0.00 | 0.00 | 0.00 | **0.12** | 0.00 |
| sweep to | | 0.00 | 0.00 | 0.00 | 0.12 | **1.00** | 0.00 | 0.00 | 0.00 | 0.04 | **0.96** |
| meat off grill | | 0.04 | 0.08 | 0.00 | **0.56** | 0.48 | 0.00 | 0.00 | 0.00 | 0.12 | **0.60** |
| turn tap | | **0.08** | 0.00 | 0.00 | 0.00 | 0.04 | 0.00 | 0.00 | 0.00 | 0.00 | **0.08** |
| reach and drag | | 0.00 | 0.00 | 0.04 | 0.28 | **0.56** | 0.08 | 0.00 | 0.00 | 0.20 | **0.44** |
| put money | | 0.08 | 0.00 | 0.16 | **0.32** | 0.28 | 0.00 | 0.00 | 0.08 | 0.08 | **0.24** |
| push buttons | | 0.08 | 0.00 | 0.00 | 0.08 | **0.48** | 0.04 | 0.00 | 0.00 | 0.04 | **0.52** |
| close jar | | 0.00 | 0.00 | 0.00 | 0.00 | **0.04** | 0.00 | 0.00 | 0.00 | 0.00 | **0.04** |
| place wine | | **0.12** | 0.00 | 0.00 | 0.00 | **0.12** | 0.04 | 0.00 | 0.00 | 0.00 | **0.08** |
| Mean S.R. | | 0.17 | 0.06 | 0.11 | 0.24 | **0.39** | 0.13 | 0.04 | 0.09 | 0.18 | **0.38** |

Table 13: Full results on zero-shot generalization of PVRs to vertical camera view changes.

| Tasks | | Vertical 5° | | | | Vertical 10° | | | |
|---|---|---|---|---|---|---|---|---|---|
| | | ○ R3M | ○ VC-1 | ○ MultiMAE | ○ PonderV2 | ○ R3M | ○ VC-1 | ○ MultiMAE | ○ PonderV2 |
| *ManiSkill2* | | | | | | | | | |
| PickCube | | **0.84** | 0.74 | 0.52 | 0.82 | **0.84** | 0.71 | 0.42 | 0.66 |
| StackCube | | 0.29 | 0.07 | 0.25 | **0.37** | 0.23 | 0.07 | 0.18 | **0.36** |
| TurnFaucet | | **0.09** | 0.06 | 0.01 | 0.07 | **0.11** | 0.06 | 0.08 | 0.07 |
| Peg- | Grasp | **0.69** | 0.49 | 0.61 | 0.66 | 0.51 | 0.37 | 0.48 | **0.65** |
| Insertion- | Align | 0.16 | 0.04 | 0.09 | **0.23** | 0.09 | 0.03 | 0.06 | **0.23** |
| Side | Insert | 0.01 | 0.00 | 0.00 | **0.03** | 0.00 | 0.00 | 0.00 | **0.03** |
| Excavate | | 0.00 | 0.00 | 0.11 | **0.13** | 0.00 | 0.00 | 0.12 | **0.14** |
| Hang | | 0.76 | 0.75 | **0.81** | 0.76 | 0.54 | **0.68** | 0.67 | 0.56 |
| Pour | | 0.06 | 0.02 | 0.00 | **0.09** | 0.03 | 0.01 | 0.00 | **0.06** |
| Fill | | 0.38 | 0.14 | 0.69 | **0.80** | 0.47 | 0.22 | **0.70** | 0.42 |
| *RLBench* | | | | | | | | | |
| open drawer | | **0.04** | 0.00 | 0.00 | 0.00 | 0.00 | 0.00 | 0.00 | 0.00 |
| sweep to dustpan of size | | 0.00 | 0.04 | 0.28 | **0.36** | 0.00 | 0.00 | 0.00 | **0.16** |
| meat off grill | | 0.08 | 0.04 | 0.00 | **0.20** | 0.04 | 0.00 | 0.00 | **0.08** |
| turn tap | | **0.04** | 0.00 | 0.00 | 0.00 | 0.00 | 0.00 | 0.00 | **0.08** |
| reach and drag | | 0.12 | 0.00 | 0.12 | **0.24** | **0.16** | 0.00 | 0.04 | 0.00 |
| put money in safe | | 0.00 | 0.00 | 0.44 | **0.48** | 0.00 | 0.00 | 0.16 | **0.28** |
| push buttons | | 0.00 | 0.00 | 0.04 | **0.04** | 0.00 | 0.00 | 0.00 | 0.00 |
| close jar | | 0.00 | 0.00 | 0.00 | **0.04** | 0.00 | 0.00 | 0.00 | 0.00 |
| place wine at rack location | | 0.04 | 0.00 | 0.00 | **0.12** | 0.04 | 0.00 | 0.00 | **0.12** |
| Mean Success rate | | 0.19 | 0.13 | 0.21 | **0.29** | 0.16 | 0.11 | 0.15 | **0.21** |

Table 14: Full results on the zero-shot generalization of scratch encoders to horizontal camera view changes.

| Tasks | | ● ResNet | ● ViT | ● MultiViT | ● SpUNet | ● PointNet | ● ResNet | ● ViT | ● MultiViT | ● SpUNet | ● PointNet |
|---|---|---|---|---|---|---|---|---|---|---|---|
| | | | | Horizontal 5° | | | | | Horizontal 10° | | |
| *ManiSkill2* | | | | | | | | | | | |
| PickCube | | 0.65 | 0.01 | 0.02 | 0.70 | **0.85** | 0.55 | 0.02 | 0.03 | 0.65 | **0.84** |
| StackCube | | 0.25 | 0.00 | 0.00 | 0.23 | **0.37** | 0.01 | 0.00 | 0.00 | 0.09 | **0.35** |
| TurnFaucet | | **0.14** | 0.09 | 0.13 | 0.06 | 0.00 | 0.13 | 0.09 | **0.14** | 0.08 | 0.00 |
| Peg- | Grasp | 0.52 | 0.32 | 0.15 | **0.81** | 0.767 | 0.28 | 0.17 | 0.15 | **0.81** | 0.77 |
| Insertion- | Align | 0.11 | 0.04 | 0.00 | 0.27 | **0.39** | 0.30 | 0.02 | 0.01 | 0.28 | **0.40** |
| Side | Insert | 0.00 | 0.00 | 0.00 | 0.01 | **0.01** | 0.00 | 0.00 | 0.00 | **0.01** | 0.00 |
| Excavate | | 0.00 | 0.00 | 0.00 | 0.00 | **0.01** | 0.00 | 0.00 | 0.00 | 0.00 | **0.02** |
| Hang | | 0.81 | 0.76 | 0.47 | 0.79 | **0.93** | 0.74 | 0.33 | 0.45 | 0.71 | **0.92** |
| Pour | | 0.04 | 0.00 | 0.00 | 0.09 | **0.13** | 0.01 | 0.00 | 0.00 | 0.05 | **0.11** |
| Fill | | 0.30 | 0.08 | 0.74 | 0.03 | **0.90** | 0.31 | 0.88 | **0.74** | 0.02 | 0.84 |
| *RLBench* | | | | | | | | | | | |
| open drawer | | 0.00 | 0.00 | 0.00 | 0.00 | 0.00 | 0.00 | 0.00 | 0.00 | 0.00 | 0.00 |
| sweep to | | 0.24 | 0.20 | 0.04 | 0.80 | **1.00** | 0.00 | 0.00 | 0.08 | 0.08 | **0.96** |
| meat off grill | | 0.16 | 0.12 | 0.00 | 0.48 | **0.56** | 0.04 | 0.20 | 0.00 | 0.32 | **0.72** |
| turn tap | | 0.00 | 0.00 | 0.00 | 0.00 | **0.04** | **0.04** | 0.04 | 0.00 | 0.00 | 0.00 |
| reach and drag | | 0.68 | 0.36 | 0.04 | 0.04 | **0.88** | 0.32 | 0.04 | 0.00 | 0.00 | **0.80** |
| put money | | **0.60** | 0.32 | 0.16 | 0.36 | 0.52 | 0.24 | 0.20 | 0.00 | 0.24 | **0.60** |
| push buttons | | 0.28 | 0.16 | 0.08 | 0.04 | **0.48** | 0.12 | 0.08 | 0.04 | 0.00 | **0.32** |
| close jar | | 0.00 | 0.00 | 0.04 | 0.00 | **0.08** | 0.00 | 0.00 | 0.00 | 0.00 | 0.00 |
| place wine | | 0.04 | 0.00 | 0.00 | 0.00 | **0.28** | **0.08** | 0.00 | 0.00 | 0.00 | **0.20** |
| Mean S.R. | | 0.25 | 0.13 | 0.10 | 0.25 | **0.43** | 0.17 | 0.11 | 0.09 | 0.17 | **0.41** |

Table 15: Full results on zero-shot generalization of PVRs to horizontal camera view changes.

| Tasks | | ○ R3M | ○ VC-1 | ○ MultiMAE | ○ PonderV2 | ○ R3M | ○ VC-1 | ○ MultiMAE | ○ PonderV2 |
|---|---|---|---|---|---|---|---|---|---|
| | | | Horizontal 5° | | | | Horizontal 10° | | |
| *ManiSkill2* | | | | | | | | | |
| PickCube | | 0.82 | 0.72 | 0.49 | **0.85** | 0.80 | 0.62 | 0.44 | **0.84** |
| StackCube | | **0.28** | 0.05 | 0.20 | 0.23 | 0.00 | 0.04 | **0.17** | 0.09 |
| TurnFaucet | | 0.09 | 0.08 | **0.11** | 0.07 | 0.09 | **0.70** | 0.13 | 0.06 |
| Peg- | Grasp | 0.15 | 0.18 | 0.28 | **0.65** | 0.56 | 0.43 | 0.55 | **0.65** |
| Insertion- | Align | 0.00 | 0.00 | 0.00 | **0.02** | 0.11 | 0.06 | 0.10 | **0.22** |
| Side | Insert | 0.11 | 0.06 | 0.10 | **0.22** | 0.00 | 0.00 | 0.00 | **0.02** |
| Excavate | | 0.00 | 0.00 | **0.11** | 0.03 | 0.00 | 0.00 | **0.11** | 0.07 |
| Hang | | **0.84** | 0.78 | 0.45 | 0.77 | 0.80 | 0.67 | 0.46 | **0.72** |
| Pour | | 0.04 | 0.03 | 0.00 | **0.09** | 0.02 | 0.01 | 0.00 | **0.05** |
| Fill | | 0.46 | 0.11 | **0.70** | 0.03 | 0.02 | 0.09 | **0.70** | 0.02 |
| *RLBench* | | | | | | | | | |
| open drawer | | 0.00 | 0.00 | 0.00 | 0.08 | 0.00 | 0.04 | 0.04 | **0.04** |
| sweep to dustpan of size | | 0.28 | 0.24 | 0.56 | **0.72** | 0.00 | 0.04 | **0.24** | 0.04 |
| meat off grill | | 0.16 | 0.20 | 0.04 | **0.76** | 0.04 | 0.16 | 0.00 | **0.64** |
| turn tap | | 0.00 | 0.00 | 0.00 | **0.08** | 0.00 | 0.00 | 0.04 | **0.04** |
| reach and drag | | 0.00 | 0.16 | **0.60** | 0.36 | 0.00 | 0.00 | 0.04 | **0.32** |
| put money in safe | | 0.04 | 0.32 | **0.40** | 0.32 | 0.04 | 0.16 | **0.16** | 0.04 |
| push buttons | | 0.08 | **0.16** | 0.08 | 0.04 | 0.04 | **0.12** | 0.04 | 0.08 |
| close jar | | 0.00 | 0.04 | **0.20** | 0.12 | 0.00 | 0.00 | 0.08 | **0.16** |
| place wine at rack location | | 0.00 | 0.00 | 0.00 | **0.12** | 0.00 | 0.00 | 0.04 | **0.08** |
| Mean S.R. | | 0.18 | 0.16 | 0.23 | **0.29** | 0.13 | 0.17 | 0.18 | **0.22** |

Table 16: Full results on visual change generalization capibilities.

| | ● ResNet | ● ViT | ● MultiViT | ● SpUNet | ● PointNet | ○ R3M | ○ VC-1 | ○ MultiMAE | ○ PonderV2 |
|---|---|---|---|---|---|---|---|---|---|
| *lighting intensity* | | | | | | | | | |
| 0.03 | 0.000 | 0.000 | 0.000 | **0.005** | 0.000 | 0.000 | 0.010 | **0.098** | 0.000 |
| 0.05 | 0.000 | 0.000 | 0.000 | **0.015** | 0.000 | 0.00 | 0.023 | **0.108** | 0.003 |
| 0.15 | 0.000 | 0.000 | 0.000 | **0.038** | 0.000 | 0.003 | 0.055 | **0.210** | 0.058 |
| 0.60 | 0.000 | 0.000 | 0.000 | **0.208** | 0.000 | 0.003 | 0.073 | 0.218 | **0.253** |
| 1.80 | 0.000 | 0.000 | 0.000 | **0.133** | 0.000 | 0.000 | 0.015 | 0.083 | **0.093** |
| 3.00 | 0.000 | 0.000 | 0.000 | **0.125** | 0.000 | 0.000 | 0.000 | 0.035 | **0.075** |
| *noise level* | | | | | | | | | |
| 2 | 0.008 | 0.000 | 0.000 | **0.098** | 0.000 | 0.003 | 0.000 | **0.203** | 0.163 |
| 16 | 0.008 | 0.000 | 0.000 | **0.120** | 0.000 | 0.008 | 0.000 | **0.215** | 0.148 |
| 32 | 0.008 | 0.000 | 0.000 | **0.098** | 0.000 | 0.008 | 0.050 | **0.225** | 0.138 |
| 64 | 0.008 | 0.000 | 0.000 | **0.085** | 0.000 | 0.002 | 0.005 | **0.210** | 0.150 |
| *background color* | | | | | | | | | |
| R0.2 | 0.000 | 0.000 | 0.000 | 0.205 | **0.345** | 0.000 | 0.008 | 0.038 | **0.365** |
| R0.6 | 0.000 | 0.000 | 0.000 | 0.213 | **0.360** | 0.000 | 0.008 | 0.000 | **0.330** |
| R1.0 | 0.000 | 0.000 | 0.000 | 0.218 | **0.350** | 0.000 | 0.003 | 0.000 | **0.368** |
| G0.2 | 0.000 | 0.000 | 0.000 | 0.223 | **0.353** | 0.000 | 0.025 | 0.105 | **0.340** |
| G0.6 | 0.000 | 0.000 | 0.000 | 0.213 | **0.350** | 0.000 | 0.000 | 0.003 | **0.343** |
| G1.0 | 0.000 | 0.000 | 0.000 | 0.218 | **0.3675** | 0.000 | 0.003 | 0.000 | **0.343** |

Table 17: Full results on sample efficiency.

| | sample 25% | | | | | sample 10% | | | | |
|---|---|---|---|---|---|---|---|---|---|---|
| Tasks | ● ResNet | ● ViT | ● MultiViT | ● SpUNet | ● PointNet | ● ResNet | ● ViT | ● MultiViT | ● SpUNet | ● PointNet |
| open drawer | 0.08 | 0.00 | **0.20** | 0.12 | 0.00 | 0.00 | 0.00 | **0.04** | 0.00 | 0.00 |
| sweep to | 0.00 | **0.24** | 0.04 | 0.04 | 0.00 | 0.00 | 0.00 | 0.00 | 0.00 | 0.00 |
| meat off grill | 0.00 | 0.00 | 0.00 | 0.00 | 0.00 | 0.00 | 0.00 | 0.00 | 0.00 | 0.00 |
| turn tap | 0.00 | 0.00 | 0.00 | 0.00 | 0.04 | 0.08 | 0.00 | 0.00 | 0.04 | 0.00 |
| reach and drag | 0.00 | 0.00 | 0.00 | 0.00 | 0.04 | 0.00 | 0.00 | 0.00 | 0.00 | 0.00 |
| put money | 0.00 | 0.00 | 0.00 | 0.00 | 0.00 | 0.00 | 0.00 | 0.00 | 0.00 | 0.00 |
| push buttons | 0.00 | 0.12 | 0.00 | 0.00 | 0.00 | 0.00 | 0.00 | 0.00 | 0.00 | 0.00 |
| close jar | 0.00 | 0.00 | 0.00 | 0.00 | 0.00 | 0.00 | 0.00 | 0.00 | 0.00 | 0.00 |
| place wine | 0.00 | 0.00 | 0.00 | 0.00 | 0.00 | 0.00 | 0.00 | 0.00 | 0.00 | 0.00 |
| Mean S.R. | 0.01 | **0.04** | 0.03 | 0.02 | 0.01 | **0.01** | 0.00 | 0.00 | 0.00 | 0.00 |
| Mean Rank | 1.89 | 1.56 | **1.44** | 1.56 | 1.78 | **1.11** | 1.22 | **1.11** | 1.22 | 1.22 |

| | ○ R3M | ○ VC-1 | ○ MultiMAE | ○ PonderV2 | - | ○ R3M | ○ VC-1 | ○ MultiMAE | ○ PonderV2 | - |
|---|---|---|---|---|---|---|---|---|---|---|
| open drawer | 0.04 | 0.00 | 0.00 | **0.20** | - | **0.04** | 0.00 | 0.00 | 0.00 | - |
| sweep to | **0.44** | 0.24 | 0.20 | 0.08 | - | 0.00 | 0.00 | 0.00 | 0.00 | - |
| meat off grill | 0.00 | 0.00 | 0.00 | 0.00 | - | 0.00 | 0.00 | 0.00 | 0.00 | - |
| turn tap | **0.04** | 0.00 | 0.00 | 0.00 | - | 0.00 | 0.00 | 0.00 | **0.12** | - |
| reach and drag | 0.04 | 0.00 | 0.00 | **0.04** | - | **0.04** | 0.00 | 0.00 | 0.00 | - |
| put money | 0.00 | 0.00 | 0.00 | 0.00 | - | 0.00 | 0.00 | 0.00 | 0.00 | |
| push buttons | 0.04 | **0.12** | 0.08 | 0.04 | - | 0.00 | 0.00 | 0.00 | 0.00 | - |
| close jar | 0.00 | 0.00 | 0.00 | 0.00 | - | 0.00 | 0.00 | 0.00 | 0.00 | - |
| place wine | 0.00 | 0.00 | 0.00 | 0.00 | - | 0.00 | 0.00 | 0.00 | 0.00 | - |
| Mean S.R. | **0.07** | 0.04 | 0.03 | 0.04 | - | 0.01 | 0.00 | 0.00 | **0.01** | - |
| Mean Rank | **1.33** | 1.67 | 1.89 | 1.75 | - | **1.11** | 1.33 | 1.33 | 1.22 | - |

Table 18: Full results on different design decisions.

| Policy | Input | Encoder | Samp. | Color | Coord. | Pick Cube | Stack Cube | Turn Faucet | PegInsertionSide Grasp | Align | Insert | Excavate | Hang | Pour | Fill | Mean S.R. |
|---|---|---|---|---|---|---|---|---|---|---|---|---|---|---|---|---|
| ACT | Point Cloud | ● SpUNet | | ✗ | ✓ | 0.27 | 0.01 | 0.00 | 0.66 | 0.20 | 0.01 | 0.23 | 0.72 | 0.02 | 0.41 | 0.25 |
| | | | Pre. | ✓ | ✗ | 0.21 | 0.01 | 0.00 | 0.63 | 0.13 | 0.01 | 0.27 | 0.78 | 0.01 | 0.21 | 0.22 |
| | | | | ✓ | ✓ | 0.11 | 0.03 | 0.00 | 0.65 | 0.07 | 0.00 | 0.22 | 0.79 | 0.03 | 0.16 | 0.21 |
| | | | | ✗ | ✓ | 0.03 | 0.00 | 0.00 | 0.75 | 0.27 | 0.01 | 0.16 | 0.84 | 0.00 | 0.61 | 0.27 |
| | | | Post. | ✓ | ✗ | 0.41 | 0.01 | 0.00 | 0.58 | 0.12 | 0.01 | 0.32 | 0.81 | 0.02 | 0.00 | 0.23 |
| | | | | ✓ | ✓ | 0.74 | 0.22 | 0.39 | 0.81 | 0.28 | 0.01 | 0.03 | 0.84 | 0.10 | 0.66 | **0.41** |
| | | ● PointNet | | ✗ | ✓ | 0.48 | 0.00 | 0.00 | 0.53 | 0.06 | 0.00 | 0.30 | 0.76 | 0.00 | 0.82 | 0.29 |
| | | | Pre. | ✓ | ✗ | 0.04 | 0.00 | 0.00 | 0.42 | 0.03 | 0.00 | 0.28 | 0.72 | 0.05 | 0.01 | 0.15 |
| | | | | ✓ | ✓ | 0.47 | 0.03 | 0.00 | 0.56 | 0.12 | 0.00 | 0.26 | 0.82 | 0.11 | 0.79 | 0.31 |
| | | | | ✗ | ✓ | 0.84 | 0.01 | 0.00 | 0.73 | 0.25 | 0.00 | 0.19 | 0.84 | 0.00 | 0.89 | 0.38 |
| | | | Post. | ✓ | ✗ | 0.08 | 0.00 | 0.00 | 0.42 | 0.04 | 0.00 | 0.24 | 0.81 | 0.06 | 0.57 | 0.22 |
| | | | | ✓ | ✓ | 0.84 | 0.35 | 0.00 | 0.77 | 0.40 | 0.01 | 0.27 | 0.83 | 0.14 | 0.91 | **0.45** |
| | Pointmap | ● ResNet | N/A | ✓ | Depth | 0.75 | 0.32 | 0.47 | 0.70 | 0.13 | 0.01 | 0.28 | 0.81 | 0.05 | 0.83 | 0.43 |
| | | ● ViT | N/A | ✓ | Depth | 0.15 | 0.00 | 0.38 | 0.59 | 0.07 | 0.00 | 0.04 | 0.83 | 0.00 | 0.78 | 0.28 |
| Diffusion | Point Cloud | ● SpUNet | | ✗ | ✓ | 0.03 | 0.00 | 0.29 | 0.47 | 0.04 | 0.00 | 0.09 | 0.52 | 0.00 | 0.09 | 0.15 |
| | | | Pre. | ✓ | ✗ | 0.05 | 0.00 | 0.30 | 0.55 | 0.04 | 0.00 | 0.21 | 0.68 | 0.00 | 0.14 | 0.20 |
| | | | | ✓ | ✓ | 0.10 | 0.08 | 0.25 | 0.56 | 0.03 | 0.01 | 0.13 | 0.71 | 0.00 | 0.04 | 0.19 |
| | | | | ✗ | ✓ | 0.71 | 0.04 | 0.32 | 0.82 | 0.09 | 0.00 | 0.17 | 0.67 | 0.00 | 0.21 | 0.30 |
| | | | Post. | ✓ | ✗ | 0.31 | 0.02 | 0.31 | 0.68 | 0.06 | 0.00 | 0.27 | 0.70 | 0.03 | 0.23 | 0.26 |
| | | | | ✓ | ✓ | 0.74 | 0.22 | 0.39 | 0.81 | 0.28 | 0.01 | 0.11 | 0.80 | 0.10 | 0.66 | **0.41** |
| | | ● PointNet | | ✗ | ✓ | 0.61 | 0.02 | 0.31 | 0.68 | 0.05 | 0.00 | 0.12 | 0.70 | 0.00 | 0.41 | 0.29 |
| | | | Pre. | ✓ | ✗ | 0.17 | 0.01 | 0.18 | 0.31 | 0.02 | 0.00 | 0.14 | 0.61 | 0.01 | 0.18 | 0.16 |
| | | | | ✓ | ✓ | 0.80 | 0.32 | 0.37 | 0.81 | 0.16 | 0.01 | 0.22 | 0.70 | 0.06 | 0.42 | 0.39 |
| | | | | ✗ | ✓ | 0.70 | 0.00 | 0.36 | 0.83 | 0.16 | 0.01 | 0.24 | 0.72 | 0.00 | 0.68 | 0.37 |
| | | | Post. | ✓ | ✗ | 0.19 | 0.03 | 0.22 | 0.38 | 0.03 | 0.00 | 0.18 | 0.62 | 0.04 | 0.09 | 0.18 |
| | | | | ✓ | ✓ | 0.90 | 0.24 | 0.37 | 0.87 | 0.29 | 0.01 | 0.28 | 0.78 | 0.13 | 0.69 | **0.45** |
| | Pointmap | ● ResNet | N/A | ✓ | Depth | 0.00 | 0.00 | 0.26 | 0.92 | 0.07 | 0.00 | 0.04 | 0.74 | 0.02 | 0.76 | 0.28 |
| | | ● ViT | N/A | ✓ | Depth | 0.01 | 0.00 | 0.22 | 0.80 | 0.05 | 0.00 | 0.03 | 0.53 | 0.00 | 0.04 | 0.17 |

Table 1: Different observation spaces on same encoder architectures with ACT policy.

| Tasks | | ● PointNet RGB | RGB-D | PCD | ● SpUNet RGB | RGB-D | PCD |
|---|---|---|---|---|---|---|---|
| PickCube | | 0.680 | 0.195 | **0.843** | 0.352 | 0.282 | **0.740** |
| StackCube | | 0.252 | 0.093 | **0.348** | 0.068 | 0.052 | **0.220** |
| TurnFaucet | | **0.393** | 0.335 | 0.000 | **0.447** | 0.268 | 0.388 |
| Peg- | Grasp | 0.675 | 0.565 | **0.765** | 0.658 | 0.668 | **0.810** |
| Insertion- | Align | 0.308 | 0.095 | **0.395** | 0.220 | 0.132 | **0.280** |
| Side | Insert | **0.027** | 0.000 | 0.005 | 0.007 | **0.013** | 0.008 |
| Excavate | | 0.035 | 0.038 | **0.268** | 0.058 | **0.097** | 0.032 |
| Hang | | 0.820 | **0.853** | 0.828 | 0.728 | 0.817 | **0.840** |
| Pour | | 0.112 | 0.110 | **0.135** | 0.090 | 0.000 | **0.095** |
| Fill | | 0.615 | 0.147 | **0.905** | 0.842 | 0.827 | 0.660 |
| Mean S.R. | | 0.392 | 0.243 | **0.449** | 0.347 | 0.316 | **0.407** |

Table 2: Different observation spaces on same encoder architectures with diffusion policy.

| Tasks | | • PointNet | | | • SpUNet | | |
|---|---|---|---|---|---|---|---|
| | | RGB | RGB-D | PCD | RGB | RGB-D | PCD |
| PickCube | | 0.863 | 0.430 | **0.900** | 0.623 | 0.567 | **0.710** |
| StackCube | | **0.495** | 0.140 | 0.238 | **0.143** | 0.120 | 0.035 |
| TurnFaucet | | 0.133 | 0.298 | **0.365** | 0.308 | 0.308 | **0.318** |
| Peg- | Grasp | 0.855 | 0.550 | **0.868** | 0.805 | 0.807 | **0.815** |
| Insertion- | Align | 0.243 | 0.072 | **0.285** | **0.143** | 0.072 | 0.093 |
| Side | Insert | **0.013** | 0.000 | 0.008 | 0.003 | **0.005** | 0.000 |
| Excavate | | 0.195 | 0.235 | **0.278** | 0.060 | 0.100 | **0.168** |
| Hang | | 0.740 | 0.712 | **0.778** | 0.540 | 0.600 | **0.673** |
| Pour | | 0.095 | 0.040 | **0.125** | **0.038** | 0.005 | 0.000 |
| Fill | | 0.252 | 0.165 | **0.693** | 0.203 | 0.195 | **0.208** |
| Mean S.R. | | 0.388 | 0.264 | **0.454** | 0.286 | 0.278 | **0.302** |

Table 1: Comparison of different point cloud frames.

| Tasks | | ACT Policy | | | | Diffusion Policy | | | |
|---|---|---|---|---|---|---|---|---|---|
| | | • PointNet | | • SpUNet | | • PointNet | | • SpUNet | |
| | | World | EE | World | EE | World | EE | World | EE |
| PickCube | | 0.843 | **0.915** | **0.740** | 0.540 | 0.900 | **0.935** | 0.740 | **0.842** |
| StackCube | | 0.348 | **0.442** | 0.220 | **0.465** | **0.238** | 0.145 | 0.220 | **0.370** |
| TurnFaucet | | 0.000 | 0.000 | **0.388** | 0.000 | 0.365 | **0.545** | 0.388 | **0.435** |
| Peg- | Grasp | **0.765** | 0.647 | 0.810 | **0.873** | 0.868 | **0.890** | 0.810 | **0.910** |
| Insertion- | Align | **0.395** | 0.195 | **0.280** | 0.245 | **0.285** | 0.140 | **0.280** | 0.175 |
| Side | Insert | 0.005 | **0.007** | 0.008 | **0.015** | **0.008** | 0.001 | **0.008** | 0.000 |
| Excavate | | **0.268** | 0.005 | **0.032** | 0.025 | 0.278 | **0.278** | 0.113 | **0.245** |
| Hang | | **0.828** | 0.810 | **0.840** | 0.825 | 0.778 | **0.820** | **0.798** | 0.770 |
| Pour | | 0.135 | **0.172** | 0.095 | **0.096** | 0.125 | **0.140** | **0.095** | 0.000 |
| Fill | | 0.905 | **0.923** | **0.660** | 0.558 | 0.693 | **0.740** | **0.660** | 0.068 |

