# OpenReview forum: "Point Cloud Matters: Rethinking the Impact of Different Observation Spaces on Robot Learning"
_NeurIPS.cc/2024/Datasets_and_Benchmarks_Track — NeurIPS 2024 Track Datasets and Benchmarks Poster_

### Official Review · Reviewer_jcCF · 2024-07-11
**Well Written Paper with Interesting Approach but one Fundamental Flaw**

**Rating:** 7
**Confidence:** 3
**Correctness:** See notes in review above.
**Clarity:** See notes in review above.

**Review:**

This is a very well written and clear paper. The tasks and simulators are pulled from well cited recent algorithmic benchmarks and represent a diverse range of test cases that are well designed to test the impact of different observation spaces. The benchmarks are reported with clear and comprehensive tables with reasonable metrics. The evaluations are comprehensive and done across a range of experiment types to answer different interesting questions. The authors provide a well documented open source code repository for this benchmark that enables others to easily use and build on their work. This is a very good fit for this track.

However, the reviewer has one fundamental issue with the paper. The whole point of the paper is to show that changing the observation space alone leads to improved performance. However, the point cloud observation space uses different model architectures than the RGB or RGB-D observation spaces. As such, in the reviewer's opinion. It is impossible to know if the results are due to the change in observation space or model architecture. Maybe a slightly modified SpUNet34 or PointNet with RGB or RGB-D data would actually perform just as well. To the reviewer, this is a major weakness in the paper and fundamentally undermines the otherwise great work substantially. The reviewer would want to see changes done to ensure that the same model architectures (likely with a modified input layer(s) to accept the different observation spaces) could have their results compared. Then, the claims made by the authors could be fairly evaluated and validated.

**Strengths:**

See notes in review above.

**Additional Feedback:**

See notes in review above.

**Documentation:**

See notes in review above.

**Ethics:**

No issues.

**Limitations:**

See notes in review above.

**Opportunities For Improvement:**

See notes in review above.

**Relation To Prior Work:**

See notes in review above.

**Summary And Contributions:**

This paper introduces a benchmark comparing observation spaces (RGB, RGB-D, Point Cloud) under a range of different tasks, algorithms, and even two different simulators. Benchmarks are provided in an open source repository. Contributions are the benchmarks themselves and the open-source software infrastructure. The authors also note that this is the first benchmark designed specifically to evaluate the observation space.

---

> ### Author Rebuttal · Authors · 2024-08-17
>
> Dear reviewer,
>
> Thank you for your constructive review and for recognizing the strengths of our work. We're delighted to hear that you found our paper well-written and clear, with diverse tasks and simulators, comprehensive experiments, and a well-documented open-source codebase. We are committed to addressing your concerns and further enhancing the paper.
>
> > “As such, in the reviewer's opinion. It is impossible to know if the results are due to the change in observation space or model architecture. Maybe a slightly modified SpUNet34 or PointNet with RGB or RGB-D data would actually perform just as well.”
>
> Great suggestion! These insights can further strengthen our claims and findings, enhancing the robustness of our work. We assign each image pixel an XYZ coordinate based on its pixel uv coordinate, allowing images to be processed as "plain point clouds" by point cloud networks. In this setting, we conducted experiments using both RGB and RGB-D modalities with PointNet and SpUNet, applying both ACT and diffusion policies across all ManiSkill2 tasks. The results are shown in the attached PDF file and align very well with our claims and findings.

---

> > ### Comment · Reviewer_jcCF · 2024-08-17
> >
> > Thank you for these updated experiments. These help address my main concern. I'd encourage you to find a way to include them (and some of the other additional experiments you've done for other reviewers) in the final paper.

---

> > > ### Author Response · Authors · 2024-08-17
> > > **Thank you so much!**
> > >
> > > Dear Reviewer,
> > >
> > > Thank you so much for the update! We sincerely appreciate your constructive feedback and valuable advice!
> > >
> > > We will include all these results and additional experiments in the final paper. We are pleased that the final paper will have one more page of content allowed if accepted. If space is limited, we will include the specific details in the supplementary material.

---

### Official Review · Reviewer_sM5n · 2024-07-23
**Great study on visual observation modality, but the soundness is questionable**

**Rating:** 6
**Confidence:** 4
**Correctness:** See "Opportunities For Improvement".
**Clarity:** Yes.

**Review:**

See "Strengths" and "Opportunities For Improvement".

**Strengths:**

This paper has several notable strengths:

- The importance of visual observation modality in robot learning, despite being overlooked for an extended period, is a critical factor. This comprehensive study is both timely and crucial to the community. From this perspective, **I truly appreciate the efforts made by the authors**.

- The study covers a wide range of experiments, including performance comparisons, zero-shot generalization, sample efficiency, and design decisions, providing a holistic view of the impact of observation spaces.

- The paper offers detailed analyses of various factors affecting performance, including pre-training, sampling strategies, and feature combinations.

- The paper reveals the superiority of point cloud methods in robot learning tasks, challenging the common practice of using RGB or RGB-D inputs.

**Additional Feedback:**

N/A

**Documentation:**

Yes.

**Limitations:**

The authors have discussed some limitations in the paper.

**Opportunities For Improvement:**

- My primary concern lies with the soundness of the empirical results and findings presented in this paper. I have previously experimented with the Diffusion Policy on some ManiSkill2 tasks and observed significantly better results than those reported here. For instance, in my experiments, the Diffusion Policy achieved success rates of at least 95%+ on the "Stack Cube" task using RGB-D observations, whereas the result shown in Table 2 of this paper is about 0%. While implementation details may vary (e.g., I use a relatively small CNN as the visual encoder), **such a substantial discrepancy in performance is alarming**. If the experiments have not been conducted correctly, then the findings and discussions in this paper could potentially be less informative or even misleading.

- The coordinate frame of the point cloud is an important factor in robot learning, as referenced in [1]. However, this paper does not specify which coordinate frame was used for the point cloud baselines. It would be beneficial for the authors to address this oversight and discuss the impact of different coordinate frames on the results. Such a discussion could greatly enhance the paper's credibility and provide deeper insights into the effectiveness of point cloud observations in robot learning.

- All experiments are conducted in simulated environments. While this allows for consistent and fair comparisons, it may not fully capture the challenges of real-world robotic tasks. For example, the depth information is more noisy in the real world compared to the simulation.

- The paper claims to explore the influence of various observation spaces on robot learning but limits its investigation to behavior cloning. It's important to note that robot learning encompasses much more than just behavior cloning. Expanding the experimental framework to include other methodologies, such as reinforcement learning, could provide a more comprehensive understanding.

[1] Liu, Minghua, et al. "Frame mining: a free lunch for learning robotic manipulation from 3d point clouds." _arXiv preprint arXiv:2210.07442_ (2022).

**Relation To Prior Work:**

Yes.

**Summary And Contributions:**

This paper introduces OBSBench, a benchmark specifically designed for comparing different observation spaces (RGB, RGB-D, and point cloud) in robot learning, focusing particularly on behavior cloning across 125 contact-rich manipulation tasks.

The authors conducted extensive experiments to examine the impact of different observation modalities, the use of pre-trained visual encoders, and other design choices. They conclude that point cloud observations hold significant promise for robot learning, offering superior performance and generalization capabilities compared to other modalities. The paper suggests that future research should focus on dynamic sampling techniques, multi-modal integration, and validation in real-world scenarios.

The key contribution of this paper is the comprehensive comparison of different observation spaces within the context of behavior cloning, providing valuable insights that could guide future developments in robotic manipulation.

---

> ### Author Rebuttal · Authors · 2024-08-17
>
> Dear reviewer,
>
> Thank you for your detailed and thoughtful review, and the time taken to provide constructive feedback to strengthen our work further! We truly appreciate your insights and suggestions for improvement, and we're pleased to know you find our study timely and crucial for the community. Below, we address your concerns and comments:
>
> > “ I have previously experimented with the Diffusion Policy on some ManiSkill2 tasks and observed significantly better results than those reported here.”
>
> Thank you for bringing this to our attention. We found an inconsistent implementation when testing channel-wise stacked RGB-D methods with diffusion policy on ManiSkill2. Revised results are shown below. However, all other results, including those related to depth with ACT, RLBench, and MultiViT, are correct. Here are the updated results:
>
> | Task | ResNet50 (RGB-D) | ViT-B (RGB-D) |
> | --- | --- | --- |
> | PickCube | 0.340 | 0.582 |
> | StackCube | 0.587 | 0.025 |
> | TurnFaucet | 0.235 | 0.295 |
> | (Peg) Grasp | 0.942 | 0.680 |
> | (Peg) Align | 0.110 | 0.025 |
> | (Peg) Insert | 0.007 | 0.000 |
> | Excavate | 0.233 | 0.027 |
> | Hang | 0.765 | 0.562 |
> | Pour | 0.063 | 0.000 |
> | Fill | 0.717 | 0.025 |
> | Mean S.R. | 0.400 | 0.222 |
>
> We will update the corresponding results in the paper. We find this does not impact our conclusions. The mean success rate for RGB-D results remains lower than PointNet’s 0.454 and SpUNet’s 0.411 on ManiSkill2 tasks. Additionally, this does not affect our finding 2 in Sec. 4.2 that the depth modality cannot consistently enhance performance and generally degrades it when considering the MultiMAE encoder, ACT policy, and RLBench tasks.
>
> Regarding why our performance is lower than your implementation, we believe there might exist some differences in the implementation details. For instance, we utilized only a single view, while default ManiSkill2 environments include an additional hand camera. We also avoided augmentations and did not incorporate tricks like EMA or replacing BatchNorm with GroupNorm. These *general* techniques, although beneficial for policy performance (for example, using multiple cameras can increase the success rate by 50% ~ 200%), are outside the scope of our focus on observation space and could introduce more hyperparameters, stochasticity, and varied settings across observation spaces.
>
> > “The coordinate frame of the point cloud is an important factor…”
>
> Great suggestion! In frame mining, the authors discovered that using the end-effector (EE) frame and target frame can lead to even better results.  Since implementing the target frame can be challenging sometimes, we conducted additional experiments with the EE frame using both PointNet and SpUNet. The results are in the attached PDF. We found that the EE frame can indeed enhance point cloud performance in many cases, though this can vary depending on specific tasks and networks.
>
> > “All experiments are conducted in simulated environments. … the depth information is more noisy in the real world compared to the simulation.”
>
> Good point! As mentioned in our global response, we strive to make our project reproducible, open-source, easy to follow, and accessible to all researchers. While real-world experiments are crucial, they can be challenging to compare fairly and often unaffordable for large-scale evaluations, especially for smaller teams or researchers from fields like computer vision or NLP. We believe simulators play an essential role in providing fair and comparable evaluations. Additionally, recent studies, such as [1], have shown that conclusions from modern simulators are highly consistent with real-world settings.
>
> However, we agree that real-world experiments can significantly enhance our findings, particularly with potentially noisy point clouds. To address this, we conducted additional experiments using the open-source low-cost-robot (https://github.com/AlexanderKoch-Koch/low_cost_robot) equipped with two Intel RealSense D415 RGB-D cameras. We’ve also open-sourced our real-world codebase on GitHub (https://github.com/HaoyiZhu/RealRobot) for easy reproduction by other researchers. We designed three tasks: **Reach Cube**, **Pick Cube**, and **Fold Cloth**. The Reach and Pick tasks involve a single arm with rigid objects, while the Fold task is bimanual and involves soft-body objects. The real-world results align with our simulated experiments, further supporting our conclusions. Details can be found in our global response.
>
> [1] SimplerEnv: Simulated Manipulation Policy Evaluation Environments for Real Robot Setups. Xuanlin et al. 2024
>
> > “Expanding the experimental framework to include other methodologies, such as reinforcement learning, could provide a more comprehensive understanding.”
>
> We acknowledge that robot learning involves various approaches beyond behavior cloning; however, it remains the most prevalent and effective method in practical applications. As our primary focus is on observation spaces, we believe our implementation of two state-of-the-art policies is currently adequate. We are committed to exploring additional methods, such as reinforcement learning, in the future.

---

> > ### Comment · Reviewer_sM5n · 2024-08-26
> >
> > Thank you for your response and the updated results. I have increased my rating.

---

> > > ### Author Response · Authors · 2024-08-26
> > > **Thank you very much!**
> > >
> > > Dear Reviewer,
> > >
> > > Thank you so much for the update! We sincerely appreciate your constructive feedback and valuable advice!

---

> ### Author Response · Authors · 2024-08-22
> **Follow-up on the response**
>
> Dear reviewer,
>
> We wonder if our response answers your questions and addresses your concerns? If yes, would you kindly consider raising the score? Thanks again for your very constructive and insightful feedback!

---

> ### Author Response · Authors · 2024-08-26
> **Follow up on our rebuttal**
>
> Dear reviewer,
>
> As the discussion stage is ending soon, we wonder if our response answers your questions and addresses your concerns? If yes, would you kindly consider raising the score? Thanks again for your very constructive and insightful feedback!

---

### Official Review · Reviewer_r3vY · 2024-07-24
**Review for OBSBench**

**Rating:** 7
**Confidence:** 4
**Correctness:** Yes
**Clarity:** Yes, the paper is well written.

**Review:**

Pros
- The paper is clearly and nicely written.
- This work performs extensive evaluations for different visual encoders and observation modalities, and derives conclusive empirical findings from the experiments that are very relevant to the field of robot learning.
- This work also provides insights on how to effectively process point cloud information for robot manipulation tasks.

Cons
- This study does not introduce a novel dataset or new tasks, relying instead on existing benchmarks ManiSkill2 and RLBench. The work replays existing expert trajectories with different observation modalities and evaluated models on top of it, which makes it more of an evaluation study rather than a new benchmark or dataset.
- Since this work is fully in simulated environments, conclusions drawn from the study may not hold in real-world scenarios, as point clouds in practice can be significantly noisier than in the simulated environment.

**Strengths:**

This work performs an extensive study across observation modalities and visual encoder choices for robot manipulation tasks. They use SOTA methods for both the encoders and policies which makes the results highly relevant to robot learning researchers. The empirical findings in this paper also provides valuable insights on various design choices for processing observations across different modalities.

**Additional Feedback:**

None

**Documentation:**

Yes

**Limitations:**

Yes, the authors mention that transferring the experiment setup in the real world in a reproducible manner is an open question.

**Opportunities For Improvement:**

It would be beneficial for this work to include a few real world experiments to show if the findings drawn from the simulated experiments still hold in the real world which would make the conclusions much stronger.

Additionally, do the original ManiSkill2 and RLBench support multiple observation modalities, or is additional setup required to enable this? If extra setup is needed, the paper should include details on how these modalities are supported in OBSBench.

**Relation To Prior Work:**

Yes

**Summary And Contributions:**

The paper introduces OBSBench, a new benchmark built upon ManiSkill2 and RLBench containing 125 tasks with support for three observation modalities: RGB, RGB-D, and point cloud. This work performs extensive experiments to study the effect of different observation modalities, and shows that point-cloud based methods have superior performance than RGB or RGB-D based counterparts, indicating an explicit 3D representation is needed for manipulation tasks.

---

> ### Author Rebuttal · Authors · 2024-08-17
>
> Dear Reviewer,
>
> Thank you for your thoughtful review and valuable feedback. We're pleased to hear that you find our paper well-written, the results relevant to robot learning, and the findings valuable. We will address your concerns and questions below.
>
> > “This study does not introduce a novel dataset or new tasks...”
>
> Thanks for the question. We acknowledge that our work builds on existing simulators and tasks. Our main goal is to benchmark the effectiveness of different observation spaces. You can find details in the benchmark’s submission guidelines (https://neurips.cc/Conferences/2024/CallForDatasetsBenchmarks). Our study is clearly aligned with this scope. While it is possible to design new tasks or collect new trajectories, it extends beyond our current focus and wouldn't impact our contributions or findings. As a benchmark focused on observation spaces, we implement state-of-the-art policies and encoders, provide standard pipelines, and offer an open-source, easy-to-use codebase, which we believe is sufficient to fulfill our objectives. Our benchmark has been praised as important, user-friendly, extensive, and interesting by the other three reviewers. Reviewer jcCF specifically mentioned that our work *“is a very good fit for this track.”*
>
>  > "Do the original ManiSkill2 and RLBench support multiple observation modalities...?”
>
> Thank you for raising this question. ManiSkill2 and RLBench do support multiple observation modalities, but some additional configuration is necessary. We have documented these details in our open-source codebase (https://github.com/HaoyiZhu/PointCloudMatters). We will also include more information in the supplementary material.
>
> > “Since this work is fully in simulated environments, conclusions drawn from the study may not hold in real-world scenarios,…”
> > “It would be beneficial for this work to include a few real-world experiments to show if the findings drawn from the simulated experiments still hold in the real world which would make the conclusions much stronger.”
>
> Great point! As a benchmark work, we aim for our project to be reproducible, open-source, easy to follow, and accessible to all researchers. While real-world experiments are important, they can be difficult to compare fairly and often unaffordable for large-scale evaluations. We believe that one of the most important meanings of simulators is just for fair and comparable evaluations. Additionally, recent studies, such as [1], have shown that conclusions from modern simulators are highly consistent with real-world settings.
>
> However, we highly agree that conducting real-world experiments can significantly enhance our findings, particularly when dealing with potentially noisy point clouds. To address this, we conducted additional real-world experiments using the open-source low-cost-robot (https://github.com/AlexanderKoch-Koch/low_cost_robot) equipped with two Intel RealSense D415 RGB-D cameras. We’ve also open-sourced our real-world codebase on GitHub (https://github.com/HaoyiZhu/RealRobot) for easy reproduction by other researchers. We designed three tasks: **Reach Cube**, **Pick Cube**, and **Fold Cloth**. The Reach and Pick tasks involve a single arm with rigid objects, while the Fold task is bimanual and involves soft-body objects. The real-world results from these tasks align with our simulated experiments, further supporting our conclusions. Results can be found in our global response.
>
> [1] SimplerEnv: Simulated Manipulation Policy Evaluation Environments for Real Robot Setups. Xuanlin et al. 2024

---

> > ### Comment · Reviewer_r3vY · 2024-08-27
> >
> > Thank you for your detailed response. These have addressed my main concerns, and the additional real robot experiments have shown that the findings presented in the paper are valid in a real-world robot setup. Therefore, I have raised my score from 5 to 7.

---

> > > ### Author Response · Authors · 2024-08-28
> > > **Thank you so much**
> > >
> > > Dear Reviewer,
> > >
> > > Thank you so much for the update! We sincerely appreciate your constructive feedback and valuable advice!

---

> ### Author Response · Authors · 2024-08-22
> **Follow-up on the response**
>
> Dear reviewer,
>
> We wonder if our response answers your questions and addresses your concerns? If yes, would you kindly consider raising the score? Thanks again for your very constructive and insightful feedback!

---

> ### Author Response · Authors · 2024-08-26
> **Follow up on our rebuttal**
>
> Dear reviewer,
>
> As the discussion stage is ending soon, we wonder if our response answers your questions and addresses your concerns? If yes, would you kindly consider raising the score? Thanks again for your very constructive and insightful feedback!

---

### Official Review · Reviewer_EBqC · 2024-07-24
**Relax an issue of selecting observation domains**

**Rating:** 8
**Confidence:** 4
**Correctness:** Yes
**Clarity:** Yes

**Review:**

This paper is well-structured and easy to follow. The released dataset is introduced in user-friendly format, thus easy to utilize.

**Strengths:**

Since the methods of observation or measurement are among the most important factors in robotics, the efficiency of each observation space should be carefully considered. This paper introduces related findings and concludes that point cloud representation is more efficient than others in the aspect of robot task planning. With various datasets based on egocentric vision being proposed for robot tasks, such considerations will help set efficient research directions for future robotics task planning. Additionally, the proposed dataset uses ACT and Diffusion Policy as baselines, making it efficient for validation with other methods as well.

**Additional Feedback:**

None

**Documentation:**

Yes

**Limitations:**

Discussed above

**Opportunities For Improvement:**

The overall performance based on ACT is better than that of the Diffusion approach, even though the Diffusion-based approach is more suitable for understanding image inputs. Moreover, when using RGB and RGB-D, performance drops when depth images are concatenated as channels. These results are somewhat difficult to comprehend, highlighting the need for a clear explanation of the preprocessing steps when depth images are used as inputs to the baseline. Additionally, for the Diffusion policy, there are two baselines: CNN and transformer. Both models are specialized for image understanding, so it is necessary to validate these models to ensure a clear understanding of RGB and RGB-D observations.

**Relation To Prior Work:**

Yes

**Summary And Contributions:**

This paper presents a dataset for various observation spaces in robot learning task. Using the well-known simulators, the proposed dataset provides 125 robot tasks, pipelines and policy baseline methods.

---

> ### Author Rebuttal · Authors · 2024-08-17
>
> Dear Reviewer,
>
> Thank you for your detailed review and positive feedback! We are delighted that you found our paper well-written, our method efficient and easy to follow, and our topic important. We appreciate your insights and will address your comments and questions below.
>
> > “The overall performance based on ACT is better than that of the Diffusion approach, even though the Diffusion-based approach is more suitable for understanding image inputs.”
>
> This is an insightful observation. Both diffusion policy and ACT tackle the policy learning problem using generative models. In the realm of image generation, diffusion-based approaches are currently very popular. However, in the context of  1-D policy learning, the policy must comprehend decision-making problems rather than image features. These image features are handled by earlier encoders like ResNet and ViT. We are not the first to discover that ACT can achieve superior performance; previous studies, such as [1][2], have also found that ACT performs better in various settings.
>
> [1] Mobile ALOHA: Learning Bimanual Mobile Manipulation with Low-Cost Whole-Body Teleoperation. Fu et al. 2024.
>
> [2] Waypoint-Based Imitation Learning for Robotic Manipulation. Lucy et al. 2024.
>
> > “Performance drops when depth images are concatenated as channels... need for a clear explanation of the preprocessing steps.”
>
> Thank you for pointing this out. For channel-wise stacked RGB-D, we use depth values in meters and stack them with RGB images. In the cases of MultiViT and MultiMAE, we adhere to the original implementation by using normalized depth values. Specifically, the depths are first masked to remove the lowest and highest 5% of values, then min-max normalized to the range [0, 1]. We observed that normalized depth values perform poorly, especially when training from scratch. This supports our assertion that explicit representation of 3D spatial information is crucial. For channel-wise stacked absolute depth using ResNet and ViT, Tab. 2 shows that RGB-D methods can sometimes yield better or even the best performance on certain tasks. However, this is not consistent across all tasks. Performance often drops significantly, so the average mean success rate of RGB-D methods is not guaranteed to be advantageous. We believe this is because depth data primarily focuses on geometry, while RGB data captures semantic information. Simply concatenating these inputs may not complement each other effectively and can increase the difficulty of learning. Therefore, an explicit spatial representation such as point clouds is crucial.
>
> > "For the Diffusion policy, there are two baselines: CNN and transformer... necessary to validate these models.”
>
> We appreciate your suggestion. We chose the UNet version of diffusion policy since it is more stable and robust as suggested in the original paper. Many following works also only use the UNet version variant such as in [3][4][5]. Also, as our focus is on different observation spaces, we believe that different policy variants do not affect our findings. To validate this more clearly, we have conducted additional validations to compare UNet and transformer versions of diffusion policy under different observation spaces on the PickCube task. As shown in the following table, the point cloud performs the best under different policy variants.
>
> | Obs. Space | Encoder | Diffusion Policy Variant | Mean S.R. |
> | --- | --- | --- | --- |
> | RGB | ResNet50 | UNet | 0.165 |
> | RGBD | ResNet50 | UNet | 0.340 |
> | Point Cloud | PointNet | UNet | 0.900 |
> | Point Cloud | SpUNet | UNet | 0.740 |
> | RGB | ResNet50 | Transformer | 0.280 |
> | RGBD | ResNet50 | Transformer | 0.340 |
> | Point Cloud | PointNet | Transformer | 0.915 |
> | Point Cloud | SpUNet | Transformer | 0.527 |
>
> [3] Consistency Policy: Accelerated Visuomotor Policies via Consistency Distillation. Prasad et al. 2024.
>
> [4] 3D Diffusion Policy: Generalizable Visuomotor Policy Learning via Simple 3D Representations. Yanjie et al. 2024.
>
> [5] DexCap: Scalable and Portable Mocap Data Collection System for Dexterous Manipulation. Wang et al. 2024.

---

> ### Author Response · Authors · 2024-08-22
> **Follow-up on the response**
>
> Dear reviewer,
>
> We wonder if our response answers your questions and addresses your concerns? If yes, would you kindly consider raising the score? Thanks again for your very constructive and insightful feedback!

---

### Author Rebuttal · Authors · 2024-08-17

# Global Response:
## Thank you to all reviewers and meta-reviewers! Real-world experiments have been added.

Dear reviewers and meta-reviewers,

We are grateful for all the time you've dedicated to providing us with constructive feedback and valuable advice to improve our paper. We've received four detailed and thoughtful reviews. We appreciate that all reviewers found our benchmark extensive, our experiments insightful, and our paper well-organized and clear. Reviewers also mentioned that our benchmark is user-friendly, our topic is relevant to robot learning, our study is timely and crucial, and our open-source efforts are useful.

For this rebuttal, we've conducted additional experiments, ablations, and analyses to provide more insight and address concerns. In our responses to each reviewer below, we address your individual questions and comments. The paper and supplementary PDFs will be updated with the suggested revisions.

We noticed that both reviewer r3vY and reviewer sM5n suggested that real-world experiments could make our findings more solid and convincing, particularly given the potential noise in real-world depths and point clouds.

As a benchmark work, our goal is to ensure our project is reproducible, open-source, easy to follow, and accessible to all researchers. In addition, some recent literature such as [1] indicates that the experimental conclusions in modern simulation environments are consistent with those in the real world. Thus we believe simulators can play a crucial role in providing fair and comparable evaluations.

Besides, we agree with the reviewers on the importance of real-world validation. To address this, we conducted additional real-world experiments using the open-sourced low-cost-robot (https://github.com/AlexanderKoch-Koch/low_cost_robot) equipped with two Intel RealSense D415 RGB-D cameras. Our bimanual setups (including 2 leader arms, 2 follower arms, 2 cameras, etc.) cost about $2000 in total, making them affordable and easy to replicate for researchers. We’ve also open-sourced our **real-world codebase** on GitHub (https://github.com/HaoyiZhu/RealRobot) for easy reproduction by other researchers.

We designed three tasks, and our workstation and task visualizations are shown in the attached PDF file:

- **Reach Cube:** A single arm with rigid objects. The robot is required to reach towards a cube and touch it.
- **Pick Cube:** A single arm with rigid objects. The robot is required to pick up the cube and hold it in the air.
- **Fold Cloth:** Two dual arms with soft-body objects. The robot is required to simultaneously catch one side of the cloth with two grippers and then fold the cloth in half.

We've implemented three modalities using ResNet50 encoders for RGB and RGB-D images, and PointNet for point cloud inputs. The ACT policy is employed across all experiments. All modalities share the same settings and hyperparameters. For data collection, leader arms are used to teleoperate the follower arms. Two RealSense RGB-D cameras capture the visual observations. Training is conducted on a single NVIDIA A100 GPU with a learning rate of 5e-5.

- **Reach Cube:** Trained with 45 demonstrations over 100 epochs.
- **Pick Cube:** Trained with 45 demonstrations over 500 epochs.
- **Fold Cloth:** Trained with 50 demonstrations over 1000 epochs.

All models are evaluated with 20 rollouts in the same environment. We report the success rates below, and corresponding videos are available at Google Drive (https://drive.google.com/drive/folders/1UiFgHv9QUPEM2is-N10IJ47DjeYiiFEm?usp=drive_link). The real-world results from these tasks align with our simulated experiments, further supporting our conclusions.

| Task | ReachCube | PickCube | FoldCloth |
| --- | --- | --- | --- |
| RGB | 0.60 | 0.05 | 0.65 |
| RGB-D | 0.30 | 0.20 | 0.50 |
| Point Cloud | **0.80** | **0.40** | **0.80** |

We sincerely thank the efforts and feedback from all the reviewers, and we welcome any follow-up discussions!

[1] SimplerEnv: Simulated Manipulation Policy Evaluation Environments for Real Robot Setups. Xuanlin et al. 2024

---

### Author Response · Authors · 2024-08-27
**Request for Timely Reviewer Feedback**

Dear ACs,

We hope this message finds you well. As we approach the conclusion of the discussion stage, we have yet to receive feedback from two of the reviewers.

For instance, reviewer r3vY is mainly concerned about the lack of real-world experiments. In response, we have added several real-world experiments and made everything open-sourced and reproducible. We believe our revisions address the reviewers' questions and concerns comprehensively.

To ensure there is ample time for a thorough discussion, could you kindly encourage the reviewers to submit their responses soon?

Thank you very much for your support and assistance.

Best regards,

Authors of Paper #228

---

> ### Comment · Area_Chair_CKYJ · 2024-08-27
>
> I will reach out to them.

---

> > ### Author Response · Authors · 2024-08-28
> > **Thank you very much!**
> >
> > Dear AC,
> >
> > Thank you very much for your support and assistance!

---

### Decision · Program_Chairs · 2024-09-26

**Decision:**

Accept (Poster)

**Comment:**

This paper introduces a new benchmark a new benchmark for robot learning in simulation, containing 125 tasks with support for three observation modalities. Reviewers broadly agree that the paper is clearly written, the range of experiments performed (especially during the discussion period) is representative, and the initial conclusions drawn from the new benchmark are novel. The developed benchmark has the potential to be of significant interest to the learning community in robotics, and offers the necessary code base to allow others to make use of the benchmark. However, there are concerns about the lack of real-world experiments and the applicability of the benchmark. These concerns are offset by additional experiments performed during the review period; however, these results should be incorporated into the main paper. Broadly speaking, this is a fairly strong paper and reviewers’ concerns were well addressed.

Strengths:
- This work performs extensive evaluations in a number of settings, including real-world robot experiments on a low-cost platform, supporting both the initial empirical findings and the use of the benchmark by others going forward.
- The initial insight that point clouds outperform more typical approaches to robot perception is novel and of interest to the robotics community. The provided benchmark makes it more feasible for others to study this result.
- The range of test cases provided is well designed to test tasks in a variety of observation spaces.
- The work is a good fit for the Benchmarks side of the D&B track.

Weaknesses:
- Although significant additional experiments were performed in response to reviewer concerns, these experiments have not been incorporated into the main paper, and it’s not clear that it will be feasible to do so without significant rewriting. This is a major concern, as the paper already relies on very long supplemental material.
- The study does not actually introduce a new dataset or new tasks, relying instead on a set of experiments that demonstrate the feasibility of exploring observation modalities in existing benchmarks. However, the novelty of the benchmark setup and utility of the provided code base and details somewhat counteracts this.
- The paper limits its investigation to behavior cloning, and applicability claims should clearly reflect this limitation.